# ECG-R1: Protocol-Guided and Modality-Agnostic MLLM for Reliable ECG Interpretation

Jiarui Jin [1 2 3]   Haoyu Wang [2]   Xingliang Wu [4]   Xiaocheng Fang [1 2 3]   Xiang Lan [5]   Zihan Wang [1 3]
Deyun Zhang [6]   Bo Liu [1]   Yingying Zhang [7]   Xian Wu [7]   Hongyan Li [1 3]   Shenda Hong [2]

## Abstract

Electrocardiography (ECG) serves as an indispensable diagnostic tool in clinical practice, yet existing multimodal large language models (MLLMs) remain unreliable for ECG interpretation, often producing plausible but clinically incorrect analyses. To address this, we propose ECG-R1, the first reasoning ECG MLLM designed for reliable ECG interpretation via three innovations. First, we construct the interpretation corpus using *Protocol-Guided Instruction Data Generation*, grounding interpretation in measurable ECG features and monograph-defined quantitative thresholds and diagnostic logic. Second, we present a modality-decoupled architecture with *Interleaved Modality Dropout* to improve robustness and cross-modal consistency when either the ECG signal or ECG image is missing. Third, we present *Reinforcement Learning with ECG Diagnostic Evidence Rewards* to strengthen evidence-grounded ECG interpretation. Additionally, we systematically evaluate the ECG interpretation capabilities of proprietary, open-source, and medical MLLMs, and provide the first quantitative evidence that severe hallucinations are widespread, suggesting that the public should not directly trust these outputs without independent verification. Code is available at here.

## 1. Introduction

Electrocardiogram (ECG), which captures the electrical activity of the heart through surface electrodes, is the foundation for diagnosing cardiac diseases (Hong et al., 2020; Wang et al., 2026a). As the most common data in cardiology, the reliability of ECG interpretation is crucial. Incorrect interpretations can directly result in patients not receiving appropriate treatment, leading to more severe health problems or life-threatening situations (Masoudi et al., 2006; Rafie et al., 2021). In clinical practice, automated algorithms extract ECG features (e.g., R-R interval) and generate preliminary diagnostic hypotheses, which are then cross-checked and confirmed by clinicians against medical protocols and twelve-lead waveform analysis to ensure the accuracy and reliability of the interpretation and diagnosis.

Deep learning methods have made promising advancements in cardiac anomaly detection (Kiyasseh et al., 2021; Lan et al., 2022; Na et al., 2024; Jin et al., 2025c; Li et al., 2025b; Jin et al., 2025a; Wang et al., 2026b), but still lack sufficient language interpretation capabilities and model interpretability. In recent years, medical multimodal large language models (medical MLLMs) have rapidly advanced in tasks such as medical data interpretation and clinical report generation (Huang et al., 2025). However, on routine clinical ECGs, our experiments indicate that existing medical MLLMs remain markedly inadequate in diagnostic reliability in two aspects:

**Hallucination in ECG Interpretation.** Recent proprietary MLLMs like GPT-5.1, and medical MLLMs like MedGemma (Sellergren et al., 2025), often generate ECG interpretations that seemingly exhibit structural completeness and appropriate clinical terminology. However, quantitative evaluation reveals that these outputs contain numerous hallucinated statements and exhibit low diagnostic accuracy (see Section 3.3 for details). The fundamental cause lies in the scarcity of high-quality ECG interpretation data during training. Although the recently ECG-Grounding (Lan et al., 2025) dataset provides evidence-based ECG interpretation data, its interpretation is still generated by purely prompting general-purpose LLMs that rely heavily on pretrained knowledge, which can introduce clinical errors.

---

[1]School of Intelligence Science and Technology, Peking University [2]National Institute of Health Data Science, Peking University [3]State Key Laboratory of General Artificial Intelligence, Peking University [4]Tianjin Institute of Cardiology, the Second Hospital of Tianjin Medical University [5]National University of Singapore [6]HeartVoice Medical Technology [7]Jarvis Lab, Tencent. Correspondence to: Shenda Hong <hongshenda@pku.edu.cn>, Hongyan Li <leehy@pku.edu.cn>, Xian Wu <kevinxwu@tencent.com>.

*Proceedings of the 43rd International Conference on Machine Learning*, Seoul, South Korea. PMLR 306, 2026. Copyright 2026 by the author(s).

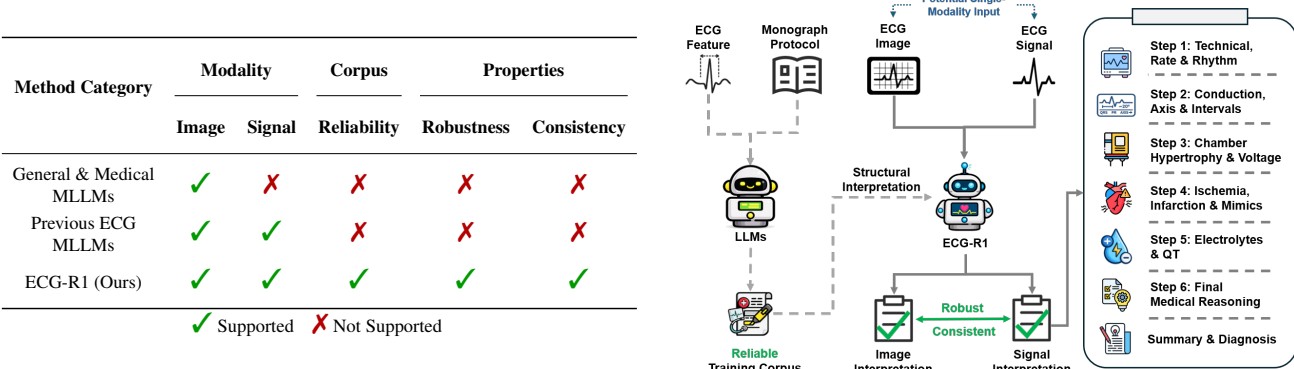

*Figure 1.* **Left:** Attribute comparison among general/medical MLLMs, previous ECG-specialized MLLMs, and ECG-R1. General/medical MLLMs typically cannot perform signal analysis and lack high-quality ECG interpretation corpora, which often leads to hallucinated, clinically incorrect interpretations at test time. Previous ECG-specialized MLLMs often construct training corpus by purely prompting LLMs from ECG features, thereby introducing medical errors that render the corpus unreliable, and they are neither robust nor cross-modal consistent under modality missing. **Right:** ECG-R1 follows a monograph-defined protocol to generate structured, clinically aligned interpretations, remaining robust and cross-modal consistent under modality missing.

**Instability and Inconsistency under Modality Missing.** As a common clinical data format, ECG is routinely presented in two forms: signals and images. Since these modalities convey consistent underlying semantic content, a model's interpretation of the same ECG should remain cross-modally consistent. However, most current medical MLLMs are limited to image-based ECG analysis and do not support signal (time-series) analysis. Although recent ECG omni-perception MLLMs like GEM (Lan et al., 2025), integrate ECG signals and ECG images, quantitative results show substantial performance degradation in metrics such as diagnostic accuracy and cross-modal inconsistency under modality missing conditions (Figure 4 and Table 2, respectively), which undermines the reliability of ECG interpretation across varying data completeness conditions.

To address the above challenges, we propose ECG-R1, the first reasoning ECG MLLM for ECG interpretation. ECG-R1 differs from previous methods in three key aspects. First, to mitigate the issue of hallucination in interpretations, we introduce a novel ***Protocol-Guided Instruction Data Generation*** paradigm. This paradigm incorporates a grounding feature extractor to extract precise physiological characteristics from ECG signals (e.g., R-R interval) and integrates a five-phase protocol extracted from a medical monograph, guiding the interpretation generation model to follow predefined thresholds and systematic diagnostic logic. Second, to address the issue of modality missing, we present a modality-decoupled architecture and introduce the theoretically motivated training strategy ***Interleaved Modality Dropout (IMD)***, which explicitly simulates random modality dropouts and interleaved inputs during training to enhance robustness and cross-modal consistency under various data completeness scenarios. Finally, as the first reasoning ECG MLLM in this field, we introduce the ***Reinforce-***

***ment Learning with ECG Diagnostic Evidence Rewards (EDER)***. Unlike general reasoning LLMs (e.g., DeepSeek-R1 (Guo et al., 2025)) that mainly optimize final-answer correctness, ECG-R1 additionally rewards structured intermediate reasoning, complementing answer-level optimization with process-level clinical reasoning and thereby improving ECG interpretation accuracy and reliability. In summary, the main contributions of this work are as follows:

- We propose ECG-R1, the first reasoning ECG MLLM for ECG interpretation, and further optimize it via reinforcement learning with ECG Diagnostic Evidence Rewards to improve ECG interpretation capability.

- We introduce Protocol-Guided Instruction Data Generation, leveraging monograph-defined thresholds and diagnostic rules to generate the corpus of structured six-step analyses with a final summary and diagnosis, while surfacing plausible unannotated abnormalities.

- We present Interleaved Modality Dropout, a theoretically motivated training strategy that simulates modality dropouts and token-block order swapping, and provide theoretical guarantees for robustness and cross-modal consistency.

- To facilitate future research, we systematically evaluate ECG interpretation across proprietary, open-source, and medical MLLMs, revealing substantial limitations in current models, and further validate ECG-R1 via licensed-cardiologist evaluation, demonstrating superior reliability and clinical usefulness.

**Conflict of Interest Disclosure.** Yingying Zhang and Xian Wu are employees of Tencent. The other authors declare no financial conflicts of interest related to this work.

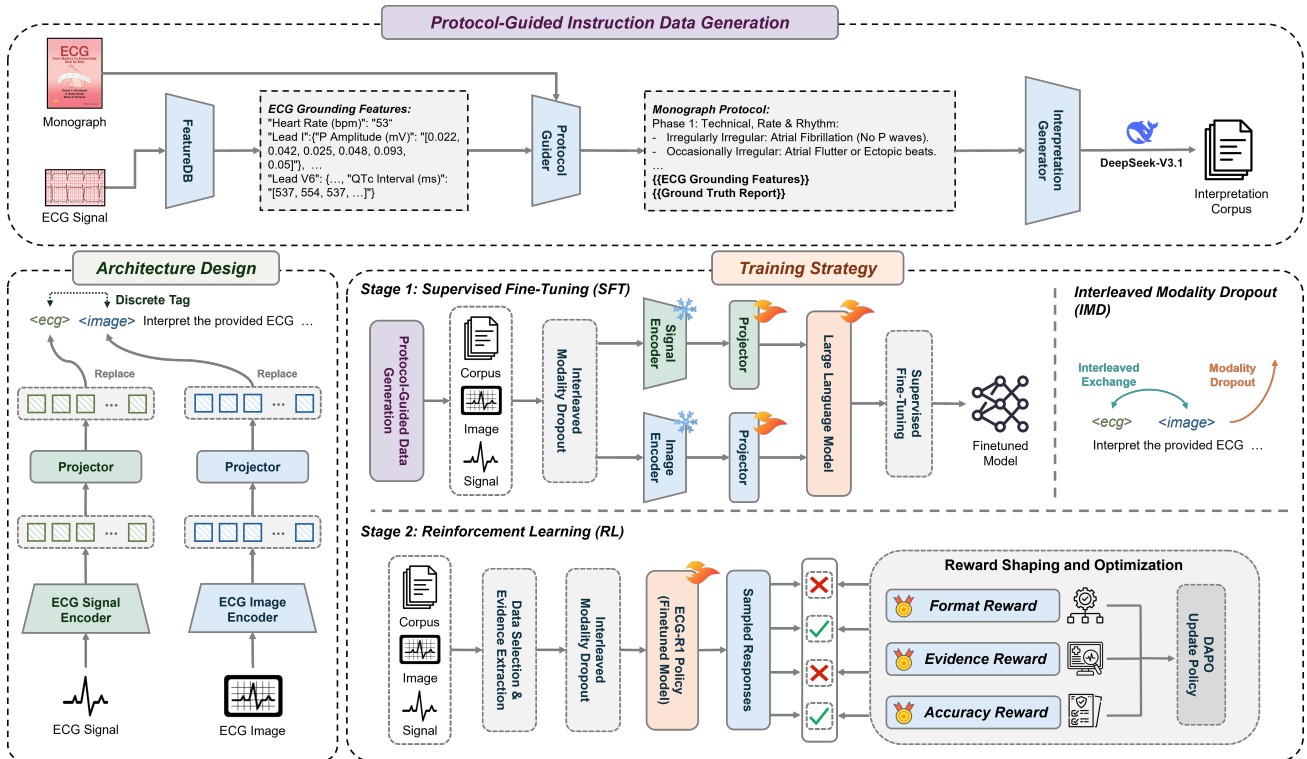

*Figure 2.* Framework of ECG-R1. Instruction generation builds a protocol-guided interpretation corpus by combining ECG grounding features with the monograph protocol. Architecture adopts a decoupled dual-encoder design with lightweight projectors to align modality-specific representations into a shared LLM space. Training follows a two-stage strategy with SFT followed by RL, and integrates IMD to enhance robustness and cross-modal consistency under modality missing.

## 2. Method

### 2.1. Framework Overview

We overview the key characteristics of ECG-R1 in Figure 1 and illustrate its end-to-end training pipeline in Figure 2. Given multimodal input $x = (x^{\text{text}}, x^I, x^T)$, where $x^{\text{text}}$ denotes the instruction text, $x^I$ the rendered ECG image, and $x^T$ the ECG time-series signal, the model generates an ECG interpretation $y$. We present ECG-R1 in three components: instruction corpus curation (Section 2.2), architecture design (Section 2.3), and the training strategy (Sections 2.4–2.6).

### 2.2. Protocol-Guided Instruction Data Generation

Instruction data underpins multimodal training and critically determines how an MLLM responds to clinical queries. Recent efforts (e.g., ECG-Grounding) generate fine-grained supervision by purely prompting general-purpose LLMs to produce causal-style interpretations from diagnosis labels and extracted signal features. However, purely prompt-based generation is largely driven by pretrained priors rather than explicit diagnostic criteria, which can introduce clinically implausible causal attributions and factual medical errors (Figure 5). To mitigate these issues, we introduce *Protocol-Guided Instruction Data Generation*, which gener-

ates supervision by adhering to standardized ECG interpretation protocols derived from a clinical monograph.

**Grounding Features Extraction.** To enable feature-grounded ECG interpretation, we extract structured physiological evidence from the raw ECG time-series. For each lead and detected heartbeat, we measure fiducial-point amplitudes, waveform morphology, and key intervals, and organize them into beat-wise, time-ordered feature sequences (e.g., $[QRS_1, \ldots, QRS_K]$ for QRS duration over $K$ beats). In practice, we construct 14 sequences per lead across 12 leads, including heart rate, RR intervals, P amplitude/duration, PR interval, QRS amplitude/duration, T amplitude/duration, ST descriptors, and QT/QTc intervals. The extraction is defined as $\boldsymbol{x}^{fs} = \text{FeatureDB}(\boldsymbol{x}^T)$, where $\boldsymbol{x}^{fs}$ is a feature dictionary and FeatureDB (Hong et al., 2019) is a deterministic, non-trainable extractor.

**Protocol-Guided Diagnosis Guider.** Given the extracted feature dictionary $\boldsymbol{x}^{fs}$, our goal is to generate a low-hallucination target response $y$ without costly expert annotations. We therefore design a protocol-guided diagnosis guider that composes a protocol-aware prompt: $\boldsymbol{x}^p = \text{ProtocolGuider}(\boldsymbol{x}^{fs}, x^{\text{protocol}})$, which steers an LLM toward evidence-based ECG interpretation. Here, $x^{\text{protocol}}$ is derived from Chapter 23 (*How to Read an ECG*) of

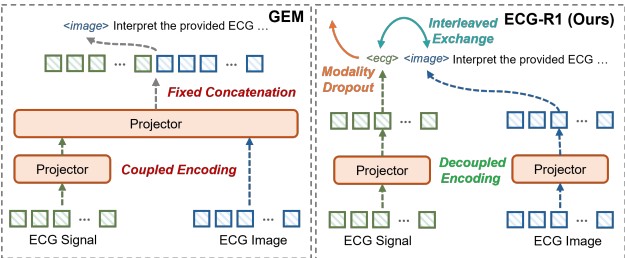

*Figure 3.* Architecture Comparison of GEM and ECG-R1.

*ECG from Basics to Essentials: Step by Step* (Stroobandt et al., 2015). The original procedure is reorganized into five phases: (i) Technical, Rate & Rhythm, (ii) Conduction, Axis & Intervals, (iii) Chamber Hypertrophy & Voltage, (iv) Ischemia, Infarction & Mimics, and (v) Electrolytes & QT. We further enforce differential exclusion with explicit negatives to rule out key mimics.

**ECG Protocol-Guided Grounding CoT Data.** Using the proposed protocol-guided generation pipeline, we curate instruction–response pairs from MIMIC-IV-ECG (Johnson et al., 2023), where each $y$ follows a fixed schema: a `<think>` block encoding the six-step protocol reasoning trace (five phases plus a final medical reasoning step), a brief summary narrative, and an `<answer>` block providing the final diagnosis. We employ DeepSeek-V3.1-Terminus (Liu et al., 2024a) as an interpretation generator. Given a protocol-structured prompt $\boldsymbol{x}^p$, it produces an interpretation $y = \text{InterpretationGenerator}(\boldsymbol{x}^p)$, where $y$ corresponds to a single protocol-guided instruction instance in our corpus. Repeating this procedure yields 30,000 protocol-guided instruction samples. Figure 5 shows a qualitative comparison. Interpretations relying on pretrained priors are prone to hallucinations and factual errors. By contrast, our protocol-guided generation injects standardized thresholds and a systematic workflow, reducing hallucination-driven mistakes and revealing clinically meaningful abnormalities missed by the original report.

### 2.3. Decoupled Modalities Encoding

**Design Motivation.** Existing ECG omni-perception MLLMs adopt a coupled encoding strategy (Figure 3): time-series (TS) embeddings are projected through a TS–Image alignment layer and then reuse the Image–Language projector before being injected into the `<image>` token space with fixed concatenation. This design (i) tightly couples TS encoding to the `<image>` placeholder, implicitly assuming paired modalities and making single-modality inference less natural and (ii) reuses a single Image–Language projector across heterogeneous modalities, creating a shared capacity bottleneck and representational compromise.

To address this issue, we instead use a *Decoupled Modalities*

*Encoding* scheme with Qwen3-VL-8B (Bai et al., 2025) (LLM + visual encoder) and ECG-CoCa (Zhao et al., 2025b) (time-series encoder). We introduce an explicit `<ecg>` tag (placed before `<image>`) and inject the time-series token block only at `<ecg>`, so the model encodes the time-series only when `<ecg>` appears. Given $x^T$ and $x^I$, we compute modality tokens and map them into the LLM embedding space via two independent projectors:

$$e^T = \text{Encoder}_T(x^T; \theta_{E_T}), \quad z^T = \text{Proj}_T(e^T; \theta_{\text{Proj}_T}),$$
$$e^I = \text{Encoder}_I(x^I; \theta_{E_I}), \quad z^I = \text{Proj}_I(e^I; \theta_{\text{Proj}_I}),$$

(1)

where $\theta_M \triangleq \{\theta_{\text{Proj}_I}, \theta_{\text{Proj}_T}\}$. and $z^T, z^I \in \mathbb{R}^{\cdot \times d}$. We condition the LLM on $x^{\text{text}}$ together with modality tokens $(z^T, z^I)$, injecting $z^T$ at `<ecg>` and $z^I$ at `<image>`.

### 2.4. Interleaved Modality Dropout

**Theoretical Motivation.** Prior ECG omni-perception MLLMs adopt a fixed fusion pattern, typically concatenating the time-series and image token blocks in a predetermined order. Training under this single canonical fusion pattern makes the model sensitive to test-time deviations (e.g., missing a modality). Such shifts degrade performance and exacerbate cross-modal inconsistency.

*Insight.* We represent common test-time deviations of ECG fusion as a finite set of environments $\mathcal{T}_{\text{test}}$ (formalized in Setup), covering modality missingness and modality-block order changes. In ECG, the image and time-series are two renderings of the same waveform. Assuming typical preprocessing where external visual annotations (e.g., cardiologist notes) are excluded, the intrinsic view asymmetry $\Delta_{\text{view}}$ is expected to be small, unlike generic omni-perception tasks where different views carry distinct information (e.g., speech for semantics; vision for spatial information).

To address these issues, we propose *Interleaved Modality Dropout (IMD)*. During supervised fine-tuning (SFT) and reinforcement learning (RL) stages, we sample a transformation $\tau \sim q$ over $\mathcal{T}_{\text{test}}$ by (i) randomly dropping either the image or the time-series modality and (ii) optionally swapping the order of their token blocks. This corresponds to minimizing a mixture risk $R_q(\theta) = \mathbb{E}_{\tau \sim q}[R_\tau(\theta)]$ over all test-relevant environments instead of a single fixed pattern, which we later show provably upper-bounds the worst-environment risk $R_{\max}(\theta)$ and, in the ECG setting where intrinsic view asymmetries $\Delta_{\text{view}}$ and $\Delta_{\text{swap}}$ are negligible, also controls the cross-modal discrepancies $\mathcal{F}(\theta)$ and $\mathcal{F}_{\text{swap}}(\theta)$, thereby yielding explicit robustness and consistency guarantees without introducing additional missing-modality generators or explicit alignment losses.

**Setup.** Let $x = (x^{\text{text}}, x^I, x^T)$ and $y$ denote the target output text. Under teacher forcing, define the NLL loss

$\ell_\theta(x,y) \triangleq -\log P_\theta(y \mid x)$. IMD samples a transformation $\tau \sim q$ from the finite environment set $\mathcal{T}_{\text{test}} = \{\tau_I, \tau_T, \tau_{IT}, \tau_{TI}\}$, where $\tau_I, \tau_T$ drop one modality and $\tau_{IT}, \tau_{TI}$ swap the two modality token blocks. In practice, $q$ is induced by two independent trials: (i) a modality-drop trial with probability $p_d$; and (ii) conditioned on retaining both modalities, a token-swap trial with probability $p_s$.

Define the environment risk $R_\tau(\theta) \triangleq \mathbb{E}_{(x,y)}[\ell_\theta(\tau(x), y)]$, the mixture risk $R_q(\theta) \triangleq \mathbb{E}_{\tau \sim q}[R_\tau(\theta)]$, and the worst-environment risk $R_{\max}(\theta) \triangleq \max_{\tau \in \mathcal{T}_{\text{test}}} R_\tau(\theta)$.

**Assumption 2.1** (Coverage). *There exists $\alpha > 0$ such that $q(\tau) \geq \alpha$ for all $\tau \in \mathcal{T}_{\text{test}}$.*

**Modality Robustness.** We study robustness to the modality-drop and token-swap transformations in $\mathcal{T}_{\text{test}}$, quantified by the worst-environment risk $R_{\max}(\theta)$.

**Theorem 2.2** (Robustness under IMD). *Under Assumption 2.1, $R_{\max}(\theta) \leq \alpha^{-1} R_q(\theta)$, where in our implementation $\alpha = \min\{p_d/2, (1-p_d)p_s, (1-p_d)(1-p_s)\}$.*

**Modality Consistency.** Define $\mathcal{F}(\theta) \triangleq \mathbb{E}_x \, \text{TV}(P_\theta(\cdot \mid \tau_I(x)), P_\theta(\cdot \mid \tau_T(x)))$ and $\mathcal{F}_{\text{swap}}(\theta) \triangleq \mathbb{E}_x \, \text{TV}(P_\theta(\cdot \mid \tau_{IT}(x)), P_\theta(\cdot \mid \tau_{TI}(x)))$. Allowing intrinsic information disparity, let $\Delta_{\text{view}} \triangleq \mathbb{E}_x \, \text{TV}(P^\star(\cdot \mid \tau_I(x)), P^\star(\cdot \mid \tau_T(x)))$ and $\Delta_{\text{swap}} \triangleq \mathbb{E}_x \, \text{TV}(P^\star(\cdot \mid \tau_{IT}(x)), P^\star(\cdot \mid \tau_{TI}(x)))$. In ECG, $\tau_I(x)$ and $\tau_T(x)$ are two renderings of the same waveform, so $\Delta_{\text{view}}$ and $\Delta_{\text{swap}}$ are negligible.

**Theorem 2.3** (Consistency via excess risk). *Let $R_\tau^\star$ be the Bayes-optimal risk in environment $\tau$ and $\varepsilon_\tau(\theta) \triangleq R_\tau(\theta) - R_\tau^\star$. Then*

$$\mathcal{F}(\theta) \leq \Delta_{view} + \sqrt{\varepsilon_{\tau_I}(\theta)/2} + \sqrt{\varepsilon_{\tau_T}(\theta)/2},$$
$$\mathcal{F}_{swap}(\theta) \leq \Delta_{swap} + \sqrt{\varepsilon_{\tau_{IT}}(\theta)/2} + \sqrt{\varepsilon_{\tau_{TI}}(\theta)/2}.$$

*Moreover, by coverage, for any $\tau \in \mathcal{T}_{\text{test}}$, $R_q(\theta) - \bar{R}_q^\star \geq \alpha \, \varepsilon_\tau(\theta)$, where $\bar{R}_q^\star \triangleq \mathbb{E}_{\tau \sim q}[R_\tau^\star]$.*

**Proofs.** See Appendix D.2 and Appendix D.3.

**Takeaway.** Theorems 2.2 and 2.3 together establish that optimizing the IMD mixture risk $R_q(\theta)$ enforces both desiderata over $\mathcal{T}_{\text{test}}$: by coverage, small $R_q(\theta)$ upper-bounds the worst-environment risk $R_{\max}(\theta)$, while the mixture excess risk $R_q(\theta) - \bar{R}_q^\star$ controls cross-modal discrepancy. In ECG, the image and time-series are two renderings of the same waveform, so $\Delta_{\text{view}}$ (and $\Delta_{\text{swap}}$) is negligible, and reducing $R_q(\theta)$ directly promotes paired-view agreement and swap invariance. In contrast, fixed concatenation minimizes only $R_{\tau_0}(\theta)$, with no guarantees on $R_{\max}(\theta)$ or cross-modal consistency in unseen test environments.

## 2.5. Supervised Fine-tuning

We conduct one-epoch supervised fine-tuning (SFT) on the instruction-tuning set $\mathcal{D}_{\text{SFT}}$ (defined in Section 3.1), which pools our protocol-guided instruction samples with a public ECG instruction dataset, and optimize only $\theta = \{\theta_M, \theta_{LLM}\}$ with IMD:

$$\min_{\theta = \{\theta_M, \theta_{LLM}\}} \mathbb{E}_{(x,y) \sim \mathcal{D}_{SFT}} \left[ -\log \pi_\theta(y \mid x) \right]. \quad (2)$$

## 2.6. Reinforcement Learning with ECG Diagnostic Evidence Rewards

After SFT, we further post-train the model with reinforcement learning (RL) to improve reasoning ability. Prior reasoning MLLMs (e.g., DeepSeek-R1) mainly reward format compliance and final-answer correctness, leaving intermediate reasoning unsupervised and thus permitting hallucinated interpretation steps. Because ECG interpretation requires stage-wise evidence grounding and differential exclusion, inspired by R1-VL (Zhang et al., 2025), we propose *ECG Diagnostic Evidence Rewards (EDER)*, which adds stepwise evidence rewards beyond outcome and format, encouraging verifiable reasoning at each step.

**Training Data.** Starting from the overall instruction corpus $\mathcal{D}_{\text{SFT}}$, we construct the RL training set $\mathcal{D}_{\text{RL}}$ as a controlled subset of the ECG Protocol-Guided Grounding CoT. Specifically, we first identify the top-500 most frequent report texts, and for each report type we retain up to 10 samples via uniform sampling without replacement (keeping all samples when fewer than 10 are available). We then apply a global shuffle with a fixed random seed of 42, resulting in $|\mathcal{D}_{\text{RL}}| = 3{,}948$ training samples.

**Extracting Key Diagnostic Evidence.** We employ DeepSeek-V3.1-Terminus as an evidence extractor. Given the reference protocol-structured $K$-step reasoning trace $y$ associated with each training instance in $\mathcal{D}_{\text{RL}}$ (with $K=6$ in our protocol), it extracts step-specific key diagnostic evidence phrases $\mathcal{E}_k(y)$ that are directly relevant to the ground-truth diagnosis labels $a^\star$. To ensure clinical specificity, we retain up to three salient evidence phrases per step, cap each phrase at six words, and prioritize diagnostically actionable cues with explicit abnormal descriptors or negative findings.

**Reward Shaping and Optimization.** Under IMD, given a training pair $(x, y) \sim \mathcal{D}_{\text{RL}}$, where $x = (x^{\text{text}}, x^I, x^T)$ and $y$ is the protocol-structured target response containing a $K$-step interpretation trace (with $K=6$ in our protocol), a brief summary narrative, and a final diagnosis $a^\star$. The policy $\pi_\theta$ samples a response $\tilde{y} \sim \pi_\theta(\cdot \mid x)$ with the same structure and a predicted diagnosis $\hat{a}(\tilde{y})$.

*Table 1.* Grounded ECG Interpretation Results.

| Metric | Diagnosis Accuracy | Analysis Completeness | Analysis Relevance | Lead Evidence Validity | ECG Feature Grounding | Evidence Based Reasoning | Clinical Diagnostic Fidelity |
|---|---|---|---|---|---|---|---|
| *Proprietary MLLMs* | | | | | | | |
| Gemini-3-Pro | 13.40 | 2.41 | 0.97 | 0.74 | 30.13 | 21.47 | 27.48 |
| GPT-5.1-Instant | 31.48 | 3.03 | 1.48 | 1.92 | 47.29 | 40.33 | 43.46 |
| *Open-source MLLMs* | | | | | | | |
| MiMo-VL-7B-SFT | 11.28 | 1.21 | 0.50 | 0.24 | 23.52 | 21.24 | 25.22 |
| GLM-4.1V-9B-Base | 17.53 | 1.94 | 0.78 | 0.57 | 30.83 | 23.97 | 25.24 |
| Qwen3-VL-8B-Instruct | 20.03 | 2.67 | 0.82 | 0.32 | 34.20 | 28.35 | 31.17 |
| InternVL3-8B-Instruct | 20.94 | 1.78 | 0.97 | 0.18 | 27.80 | 20.43 | 21.55 |
| MiniCPM-V-4.5 | 25.29 | 2.83 | 1.40 | 0.45 | 38.82 | 31.12 | 34.77 |
| *Medical MLLMs* | | | | | | | |
| MedVLM-R1 | 16.62 | 0.49 | 0.13 | 0.00 | 11.36 | 5.63 | 5.12 |
| Chiron-o1-8B | 21.20 | 1.57 | 1.01 | 0.45 | 30.81 | 24.87 | 25.88 |
| QoQ-Med-VL-7B | 27.01 | 2.56 | 1.79 | 0.42 | 37.38 | 32.85 | 33.70 |
| MedGemma-4B | 27.34 | 2.10 | 0.92 | 0.02 | 26.24 | 21.52 | 23.14 |
| MedGemma-27B | 25.23 | 3.20 | 1.50 | 0.81 | 42.04 | 36.14 | 39.22 |
| HuatuoGPT-Vision-7B | 29.27 | 1.83 | 1.34 | 0.18 | 33.21 | 29.41 | 29.85 |
| *ECG-specialized MLLMs* | | | | | | | |
| PULSE | 66.13 | 1.90 | 1.86 | 0.19 | 41.6 | 39.42 | 40.53 |
| GEM | 74.70 | 4.25 | 3.79 | 4.41 | 65.34 | 63.15 | 62.90 |
| ECG-R1 (SFT) | 79.33 | 6.36 | 4.58 | 5.53 | 79.92 | 78.08 | 83.51 |
| ECG-R1 (RL) | **80.29** | **6.51** | **4.74** | **5.81** | **80.57** | **79.08** | **84.20** |

To reward stepwise evidence grounding, we compute a per-step evidence coverage score. Let $\mathcal{E}_k(y)$ denote the set of step-specific key evidence phrases extracted from the *k-th step text* of the target response $y$, and let $\tilde{y}^{(k)}$ denote the generated text at step $k$. We define the step reward as

$$r_{\text{step}}^{(k)}(x, \tilde{y}; y) = \frac{\left| \text{match}(\mathcal{E}_k(y), \tilde{y}^{(k)}) \right|}{\left| \mathcal{E}_k(y) \right|}, \qquad (3)$$

where $\text{match}(\cdot, \cdot)$ counts the number of evidence phrases that appear in $\tilde{y}^{(k)}$ after simple normalization (e.g., case-folding). We then compute the diagnostic process reward by averaging per-step scores:

$$R_{\text{EDER}}(x, \tilde{y}; y) = \frac{1}{K} \sum_{k=1}^{K} r_{\text{step}}^{(k)}(x, \tilde{y}; y). \qquad (4)$$

The diagnosis accuracy reward is computed from the predicted diagnosis in the <answer> block as a set-level Jaccard similarity:

$$R_{\text{accuracy}}(x, \tilde{y}) = \frac{\left| \mathcal{S}(\hat{a}(\tilde{y})) \cap \mathcal{S}(a^\star) \right|}{\left| \mathcal{S}(\hat{a}(\tilde{y})) \cup \mathcal{S}(a^\star) \right|}. \qquad (5)$$

where $\mathcal{S}(\cdot)$ maps a diagnosis string to a label set by splitting on semicolons for multi-label cases and treating it as a singleton otherwise. We include a format reward $R_{\text{format}}(\tilde{y})$, which assigns 1 if $\tilde{y}$ contains a non-empty <think> block and 0 otherwise. Finally, the total reward is then defined as

$$R_{\text{total}} = R_{\text{format}} + R_{\text{accuracy}} + \lambda R_{\text{EDER}}, \qquad (6)$$

where $\lambda$ controls the weight of the stepwise evidence-grounding reward. After computing rewards, we optimize the policy with DAPO (Yu et al., 2025a). For each $x$, we sample $G$ responses $\tilde{y}_i$ from $\pi_{\theta_{\text{old}}}(\cdot \mid x)$, compute $R_i = R_{\text{total}}(x, \tilde{y}_i; y)$, and set a per-response advantage $\hat{A}_i = (R_i - \text{mean}(\{R_j\}))/\text{std}(\{R_j\})$, shared across tokens in $\tilde{y}_i$. Let $r_{i,t}(\theta) = \pi_\theta(\tilde{y}_{i,t} \mid x, \tilde{y}_{i,<t})/\pi_{\theta_{\text{old}}}(\tilde{y}_{i,t} \mid x, \tilde{y}_{i,<t})$. DAPO updates $\theta$ by maximizing the decoupled-clipping PPO objective with $N = \sum_{i=1}^{G} |\tilde{y}_i|$, $\epsilon_{\text{low}}$=0.2, and $\epsilon_{\text{high}}$=0.3: $J(\theta) = \mathbb{E}\left[ \frac{1}{N} \sum_{i,t} \min\left(r_{i,t}(\theta), \tilde{r}_{i,t}(\theta)\right) \hat{A}_i \right]$, where $\tilde{r}_{i,t}(\theta) = \text{clip}\left(r_{i,t}(\theta), 1 - \epsilon_{\text{low}}, 1 + \epsilon_{\text{high}}\right)$.

## 3. Experiments

### 3.1. Training Dataset

We define $\mathcal{D}_{\text{SFT}}$ as the union of our ECG Protocol-Guided Grounding CoT and ECGInstruct (Liu et al., 2024c). We synthesize ECG images from raw signals using ECG-image-kit (Shivashankara et al., 2024) and extract fine-grained

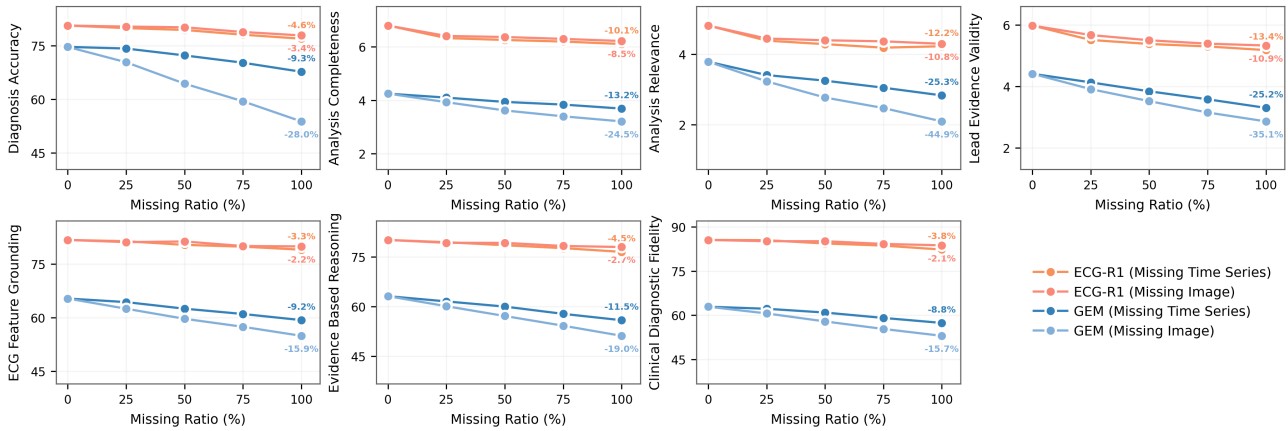

*Figure 4.* Modality Missing Results between Time-Series and Image Modalities.

features via FeatureDB.

## 3.2. Main Evaluation Tasks and Metrics

**Grounded ECG Interpretation.** We evaluate grounded ECG interpretation to assess whether the MLLM attains cardiologist-level competency in basic ECG reading, where accurate diagnosis must be accompanied by fine-grained evidence localization and clinically grounded justification. Performance is measured by seven rubric-based metrics (Appendix B.1) that jointly quantify diagnostic correctness, coverage and relevance of ECG findings, lead-wise evidence validity, and the fidelity of evidence-based clinical reasoning. For fair comparison, we adopt the ECG-Grounding test set (Lan et al., 2025) comprising 2,381 samples. Finally, to avoid dependence on an older closed-source grader (GPT-4o) with potential access restrictions and long-term unavailability, we use DeepSeek-V3.1-Terminus to score the MLLM outputs under the same evaluation rubric. To support future research, we further introduce GLM-5 as an independent verifier to re-evaluate the outputs of all models, with detailed results provided in Section C.2.

**Robust and Consistent ECG Interpretation.** We introduce the Robust and Consistent ECG Interpretation task to evaluate model performance under different levels of data completeness, focusing on robustness and consistency in modality missing conditions for ECG omni-perception MLLMs. For robustness, under the full-modality input setting, we randomly drop either the time-series or image modality, retaining only a single modality for evaluation, and assess performance using the same test set and metrics as the Grounded ECG Interpretation task. For consistency, we evaluate time-series-only and image-only inputs on the same cases and quantify cross-modal output agreement using three text-similarity metrics, whose definitions and scoring criteria are provided in Appendix B.1.

*Table 2.* Modality Consistency Results between Time-Series and Image Modalities.

| Metric | BLEU-4 | ROUGE-L | SBERT-Score |
|--------|--------|---------|-------------|
| GEM | 0.33 | 0.43 | 0.92 |
| ECG-R1 | **0.69** | **0.73** | **0.97** |

**Cardiologist Evaluation.** To assess the real-world reliability and usefulness of the interpretations, we invited four licensed cardiologists to independently review the model outputs using seven predefined clinical criteria, with detailed metric definitions and scoring rubrics provided in Appendix B.1. Specifically, we randomly sampled 100 test set cases and collected cardiologist ratings for interpretations generated by GEM (the strongest baseline) and ECG-R1, reporting results as mean and standard deviation.

## 3.3. Evaluation Results on ECG Interpretation

Table 1 summarizes the performance of representative proprietary, open-source, medical, and ECG-specialized MLLMs, together with our method, on seven evaluation metrics for ECG interpretation. Among all non-ECG-specialized MLLMs, GPT-5.1 achieves the highest Diagnosis Accuracy. However, its absolute score remains low at 31.48, falling far short of the reliability required for real-world clinical deployment. Notably, despite ECG being a routine clinical modality, existing medical MLLMs still fail to interpret ECGs reliably, with Diagnosis Accuracy consistently below 30.00. A deeper analysis reveals that most non-ECG-specialized MLLMs obtain relatively high scores in Analysis Completeness, yet perform poorly in Analysis Relevance, Lead Evidence Validity, and ECG Feature Grounding. This discrepancy indicates that these models can follow instructions to generate structurally complete and seemingly comprehensive analyses (e.g., mentioning R–R intervals), but the underlying content is often incorrect,

*Table 3.* Cardiologist Evaluation of Reliability and Usefulness Metrics (Mean and STD).

| Metric | Reliability | | | Usefulness | | | |
|---|---|---|---|---|---|---|---|
| | Analytical Relevance | Analytical Accuracy | Analytical Completeness | Reasoning Quality | Findings Novelty | Clinical Value | Overall Satisfaction |
| GPT-5.1-Instant | 2.80/5 (0.73) | 2.83/5 (0.33) | 2.75/5 (0.66) | 2.80/5 (0.73) | 2.46/5 (1.00) | 2.66/5 (0.52) | 2.66/5 (0.44) |
| PULSE | 3.59/5 (1.01) | 3.97/5 (0.10) | 3.32/5 (1.53) | 3.34/5 (1.46) | 2.94/5 (1.05) | 3.89/5 (0.17) | 3.90/5 (0.15) |
| GEM | 4.16/5 (0.78) | 3.89/5 (0.89) | 4.05/5 (0.71) | 4.03/5 (0.73) | 2.82/5 (1.59) | 3.84/5 (1.00) | 3.84/5 (0.99) |
| ECG-R1 | 4.55/5 (0.53) | 4.34/5 (0.66) | 4.43/5 (0.58) | 4.48/5 (0.57) | 3.25/5 (1.78) | 4.38/5 (0.64) | 4.38/5 (0.63) |

exhibiting systematic hallucinations that ultimately lead to erroneous diagnostic conclusions. Although MLLMs excel at general image understanding, our results provide the first systematic evidence that severe hallucination is pervasive in ECG interpretation, cautioning against uncritical public reliance on their outputs.

In comparisons with ECG-specialized MLLMs, ECG-R1 delivers consistent, comprehensive gains. Compared with GEM, the first ECG omni-perception MLLM and the previous best-performing model, ECG-R1 increases Diagnosis Accuracy to 80.29 and improves analysis quality (Analysis Completeness 6.51; Analysis Relevance 4.74), indicating interpretations that are more complete and better aligned with the final diagnosis. We attribute these gains to a structured, stepwise workflow that enforces systematic review, improving coverage of critical diagnostic elements and reducing missed clinically relevant findings. Lead Evidence Validity also rises to 5.81, suggesting more diagnosis-relevant, lead-specific evidence rather than template-style lead enumeration. Most notably, ECG-R1 yields a +17.49 average absolute gain over GEM on ECG Feature Grounding, Evidence-Based Reasoning, and Clinical Diagnostic Fidelity, indicating stronger grounding in verifiable ECG features and tighter evidence-to-diagnosis linkage. Moreover, we adopt explicit, monograph-defined phases with a fixed sequence of rhythm, conduction, morphology, and ischemia assessment, closely resembling real-world clinical ECG workflow before the final diagnosis. Finally, the RL model consistently outperforms the SFT model across all metrics, showing that evidence-rewarded reinforcement learning further strengthens reasoning performance. Collectively, these gains indicate that ECG-R1 makes substantial progress toward more reliable ECG interpretation.

### 3.4. Evaluation Results on Robustness and Consistency under Modality Missing

Figure 4 and Table 2 present the robustness and consistency evaluation results specifically designed for ECG omni-perception MLLMs, with the representative model GEM used for comparison. In the robustness evaluation, GEM exhibits substantial performance degradation under modality missing conditions. In particular, when only the time-series

modality is provided and the image modality is entirely missing, Diagnosis Accuracy score suffers a maximum relative drop of 28.0%, while Analysis Relevance score experiences an even more severe maximum relative drop of 44.9%. These results show that GEM is highly sensitive to missing modalities and lacks robustness under incomplete inputs. In contrast, ECG-R1 exhibits consistently smaller relative drops when either the time-series or image modality is removed, and even with one modality entirely absent, it still surpasses GEM using both modalities. In the consistency evaluation, we assess the agreement between interpretations generated from time-series-only and image-only inputs. GEM achieves BLEU-4 and ROUGE-L scores of 0.33 and 0.43, indicating substantial discrepancies in surface expression and content coverage across modalities. In contrast, ECG-R1 attains markedly higher scores on both metrics, demonstrating improved cross-modality textual consistency. Moreover, ECG-R1 achieves an SBERT-Score of 0.97, reflecting strong semantic alignment between interpretations produced under different modality conditions. Collectively, these results show that ECG-R1 maintains reliable ECG interpretations across varying levels of data completeness.

### 3.5. Cardiologist Evaluation Results

Table 3 reports the ratings independently provided by four licensed cardiologists. For reliability, ECG-R1 consistently outperforms GEM across all three criteria. The gains in Analytical Relevance and Analytical Completeness are driven by a structured, stepwise interpretation procedure, which keeps the analysis diagnosis-focused and systematically covers key ECG components. Crucially, Analytical Accuracy measures whether an interpretation contains medical factual errors, where lower scores indicate more frequent or more severe errors, and thus directly reflects the severity of hallucination-like failures in ECG interpretation. Under this criterion, ECG-R1 achieves higher accuracy than GEM (4.34 vs. 3.89), because ECG-R1's interpretation corpus is generated under protocol guidance, whereas GEM's interpretation corpus relies on pretrained knowledge from LLMs alone. For usefulness, ECG-R1 consistently outperforms GEM across all four criteria. The Reasoning Quality gain reflects the clinical alignment of our monograph-derived five-phase analysis, while higher Clinical Value and Over-

all Satisfaction indicate more actionable support. Findings Novelty improves but remains case-dependent with greater inter-rater variability. PULSE attains slightly higher Analytical Accuracy than GEM, likely due to fewer factual errors in its concise outputs, but shows lower Completeness and Reasoning Quality. Overall, cardiologists rate ECG-R1 interpretations as more trustworthy.

## 4. Conclusion

In this work, we propose ECG-R1, the first reasoning ECG MLLM for ECG interpretation. ECG-R1 improves reliability through three innovations: Protocol-Guided Instruction Data Generation to construct structured, comprehensive interpretations and reduce factual errors; Interleaved Modality Dropout to enhance robustness and cross-modal consistency under varying data completeness; and Reinforcement Learning with ECG Diagnostic Evidence Rewards to strengthen evidence-based reasoning. We also conduct a systematic evaluation of ECG interpretation across proprietary, open-source, and medical MLLMs, exposing key limitations in these MLLMs. Experiments show that ECG-R1 consistently outperforms prior state-of-the-art methods in diagnostic accuracy and interpretation quality, while remaining stable under modality missing conditions, representing substantial progress toward reliable ECG interpretation.

## Acknowledgements

This work was supported by the National Natural Science Foundation of China under Grant Nos.62102008 and 62172018, the CCF-Tencent Rhino-Bird Open Research Fund under Grant No.CCF-Tencent RAGR20250108, the CCF-Zhipu Large Model Innovation Fund under Grant No.CCF-Zhipu202414, the PKU-OPPO Fund under Grant Nos.BO202301 and BO202503, and the Research Project of Peking University in the State Key Laboratory of Vascular Homeostasis and Remodeling under Grant No.2025-SKLVHR-YCTS-02.

The authors would also like to thank the cardiologists who participated in the cardiologist evaluation: Yiping Wang from the Department of Cardiology, Ma'anshan 17th Metallurgical Hospital, Ma'anshan, China; Yirao Tao from the Department of Cardiology, Beijing Jishuitan Hospital, Capital Medical University, Beijing, China; Xinxin Di from the Department of Electrocardiogram, The First Affiliated Hospital of USTC, Division of Life Sciences and Medicine, University of Science and Technology of China, Hefei, Anhui, China; and Jing Zhao from the ECG & Cardiac Function Department, The First Affiliated Hospital of Anhui Medical University, Hefei, Anhui, China.

## Impact Statement

This work aims to improve the reliability of ECG interpretation produced by multimodal large language models by grounding interpretations in structured clinical logic and ECG-derived physiological measurements. If developed and validated responsibly, such approaches may help reduce hallucinated ECG interpretations produced by multimodal large language models, improve the consistency of model-generated reports under real-world data imperfections, and support research on safer AI-assisted ECG workflows.

At the same time, ECG interpretation is a high-stakes medical application, where incorrect or incomplete interpretations may affect downstream clinical assessment and patient management. Despite the improvements demonstrated in this study, the proposed methods and models are intended for research purposes and should be used only under qualified clinical oversight. Model outputs may still contain errors, omissions, or misleading statements, particularly in complex, rare, or low-quality cases. Therefore, these outputs should serve only as auxiliary references, with qualified clinicians remaining responsible for verification, interpretation, and final decision-making. Any real-world deployment would require prospective multi-site validation, clinical governance, user training, monitoring, and regulatory review.

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

# ECG-R1: Protocol-Guided and Modality-Agnostic MLLM for Reliable ECG Interpretation
## Appendix

# A. Related Work

## A.1. Medical Multimodal Large Language Model

Medical multimodal large language models (medical MLLMs) have progressed rapidly via instruction tuning for medical data understanding, clinical text generation, and cross-modal QA (Huang et al., 2025). LLaVA-Med (Li et al., 2023) utilizes PubMed Central data and GPT-4-generated instructions for multimodal tuning. HuatuoGPT-Vision (Chen et al., 2024a) leverages the PubMedVision dataset, achieving superior performance in medical VQA and report generation, especially in Chinese. MedGemma (Sellergren et al., 2025) specializes in diverse modalities like radiology and pathology, prioritizing mobile and single-GPU deployment. Recent research emphasizes reasoning and reliability. For examples, MedVLM-R1 (Pan et al., 2025) employs reinforcement learning for chain-of-thought reasoning, while Chiron-o1-8B (Sun et al., 2025) integrates MICS with tool-augmented thinking. QoQ-Med (Dai et al., 2025) reasons across images, time-series, and text using Domain-aware GRPO to mitigate clinical bias. However, despite ECG being one of the most prevalent clinical data modalities, our experiments show that existing medical MLLMs exhibit pronounced deficiencies in both the completeness and accuracy of ECG interpretation, which limits their practical applicability in real-world clinical settings.

## A.2. Language-based ECG Analysis

Language-based ECG interpretation and diagnosis remain in the early stages of exploration, with only a few recent studies discussing MLLM-based ECG analysis methods. Specifically, ECG-Chat (Zhao et al., 2025b) was developed as an MLLM focused on processing time-series ECG data for report generation. In another approach, PULSE (Liu et al., 2024c) enhances ECG image understanding for diagnosis and reporting by synthesizing realistic ECG images from raw signals. anyECG-Chat (Li et al., 2025a) serves as a multi-task MLLM capable of report generation, abnormal waveform localization, and open-ended QA. Furthermore, GEM (Lan et al., 2025) is an omni-perception MLLM that performs cross-modal fusion between time-series ECG and ECG images to produce evidence-based ECG interpretations. UniECG (Jin et al., 2025b) was also introduced as an unified model capable of both evidence-based ECG interpretation and signal generation. However, prior approaches that rely on the pretrained knowledge of general-purpose LLMs for data generation may suffer from hallucinations, thereby reducing the reliability of the resulting interpretations. Moreover, existing ECG omni-perception MLLMs still exhibit pronounced performance degradation under modality missing conditions, which hinders their prospects for real-world deployment.

# B. Experiment Details

## B.1. Metrics

**Grounded ECG Interpretation: Evaluation Metric Rubrics**

**DiagnosisAccuracy** *(0–2)*
Evaluates whether the generated diagnosis is correct, specific, and supported by ECG findings. For each sample, we treat all sub-items with $\text{Score} > 0$ as correct and report the percentage of such sub-items among all sub-items.

**+2 per diagnosis** Each correctly identified key diagnosis with supporting ECG features.

**+1 per diagnosis** Each mostly correct diagnosis but lacking key supporting details.

**+0 per diagnosis** Each incorrect or vague diagnosis not supported by ECG features.

**AnalysisCompleteness** *(0–1)*
Checks if all key ECG components (e.g., rhythm, intervals, waveforms, and lead-specific findings) are discussed. Results are provided in absolute terms, indicating the average number of correctly addressed key ECG features for each sample.

**+1 per feature** For each correctly addressed key ECG feature (e.g., rhythm, PR interval, QRS duration, ST segment, T wave morphology).

**+0 per missing feature** For each key feature omitted or inaccurately described.

**AnalysisRelevance** *(0–2)*
Assesses whether each provided explanation directly supports the diagnosis. Results showing on average how many points support the diagnosis with clear ECG evidence for each sample.

**+2 per feature or per lead** Each point that strongly supports the diagnosis with clear ECG evidence.

**+1 per feature or per lead** Some points are relevant but not fully justified.

**+0** Includes unrelated or misleading explanations.

**LeadEvidenceValidity** *(0–2)*
Evaluates whether the lead-related statements are diagnostically necessary, correctly grounded, and free of unsupported lead-wise claims, rather than maximizing the number of mentioned leads.

**+2 per key lead/region** For each diagnosis-critical lead (or contiguous lead group / territory) correctly referenced with explicit and matching ECG evidence (e.g., ST elevation/depression, T-wave inversion, pathologic Q waves, bundle-branch morphology) that supports the stated diagnosis.

**+1 per key lead/region** For each key lead/region mentioned with partially correct evidence, but missing important details (e.g., direction without magnitude/morphology) or showing minor inconsistencies.

**+0 per key lead/region** Key lead/region is omitted or the described finding is incorrect / non-grounded and does not support the diagnosis.

**GroundedECGUnderstanding** *(0–100)*
Determines if the interpretation references actual ECG features (e.g., QRS amplitude, PR interval) instead of generic terms. Results are scaled from 0 to 100.

**100 –** ECG findings are comprehensively cited, linked to diagnoses, and cover all relevant ECG features.

**80 –** ECG findings are explicitly cited and linked to diagnoses.

**50 –** Some ECG references exist but are incomplete.

**0 –** Lacks specific waveform references.

**EvidenceBasedReasoning** *(0–100)*
Evaluates whether the diagnosis follows logical, evidence-supported steps. Results are scaled from 0 to 100.

**100 –** Findings logically progress to diagnosis with thorough and clear justifications covering all necessary steps.

**80 –** Findings logically progress to diagnosis with clear justifications.

**50 –** Some reasoning exists but lacks complete step-by-step analysis.

**0 –** Reasoning is unclear or not derived from ECG findings.

**ClinicalDiagnosticFidelity** *(0–100)*

Assesses if the model mimics how a clinician interprets an ECG, considering all relevant factors. Results are scaled from 0 to 100.

**100 –** The analysis follows a structured clinical approach and considers all relevant clinical factors.

**80 –** The analysis follows a structured clinical approach.

**50 –** Some clinical reasoning is present but incomplete.

**0 –** The approach lacks structured clinical reasoning.

## Consistent ECG Interpretation Metrics

*BLEU-4* measures cross-modal n-gram fidelity between the time-series-only and image-only generated interpretations by computing the modified precision over 1–4-grams with a brevity penalty to discourage overly short outputs; higher BLEU-4 indicates stronger surface-form agreement in local phrasing and word order.

*ROUGE-L* evaluates cross-modal sequence-level overlap using the longest common subsequence (LCS) between the two generated texts, typically summarized as an F-measure over LCS-based precision and recall; higher ROUGE-L reflects better global structural consistency while allowing limited rephrasing.

*SBERT-Score* assesses cross-modal semantic agreement by encoding each generated text into a dense sentence embedding using the embedding model BGE-M3 (Chen et al., 2024b), and then computing the cosine similarity between the two embeddings, capturing meaning-level alignment beyond lexical overlap.

## Cardiologist Evaluation: Reliability Metrics

**Analytical Relevance**                                                                                    *(1–5)*
Does the model's analysis closely support the diagnosis and provide corresponding ECG evidence?

**5 –** Every analysis point is highly relevant to the diagnosis, with clear supporting evidence.

**4 –** Most analyses are strongly relevant, with minor insufficiencies.

**3 –** Some analyses are relevant, but there is clear irrelevant content.

**2 –** Most analyses are weakly relevant.

**1 –** The analysis is unrelated to the diagnosis.

**Analytical Accuracy**                                                                                      *(1–5)*
Are there any medical factual errors in the model's output?

**5 –** Completely accurate.

**4 –** Mostly accurate.

**3 –** Some errors.

**2 –** Obvious errors.

**1 –** Severe errors.

**Analytical Completeness**                                                                                  *(1–5)*
Does the model comprehensively discuss key ECG components relevant to the diagnosis, including rhythm, intervals, and waveforms?

**5 –** All relevant ECG features (rhythm, PR, QRS, ST, T waves, intervals, etc.) are accurately discussed.

**4 –** Most key ECG features are covered, with minor omissions.

**3 –** Only some features are covered, with significant gaps.

**2 –** Only a few ECG features are mentioned.

**1 –** ECG components are largely missing, with severe omissions.

## Cardiologist Evaluation: Usefulness Metrics

**Reasoning Quality**                                                                                        *(1–5)*
Does the model provide a clear, evidence-based reasoning process similar to that of a clinician, logically deriving the diagnosis from ECG features?

**5 –** Clear and coherent reasoning structure, explaining each step from ECG to diagnosis causally.

**4 –** Overall reasonable reasoning, but some steps lack detail.

**3 –** Partial reasoning present, but incomplete or logically weak.

**2 –** Disjointed reasoning with major gaps.

**1 –** No logical reasoning, only a stack of conclusions.

**Findings Novelty** *(1–5)*
Does the model provide insights or findings not noticed by the clinician?

**5 –** Important new diagnoses or findings.

**4 –** Novel and somewhat insightful content.

**3 –** Some new findings, but of limited value.

**2 –** Conventional content, not particularly insightful.

**1 –** No new information.

**Clinical Value** *(1–5)*
Does the model output help in clinical decision-making?

**5 –** Direct and significant support for clinical judgment; content is clear and reliable.

**4 –** Most content is helpful and practically useful.

**3 –** Somewhat informative, but basic or unclear.

**2 –** Partially suggestive, with limited decision support.

**1 –** No value for clinical judgment; not informative.

**Overall Satisfaction** *(1–5)*
Subjective rating of the overall output quality.

**5 –** Very satisfied.

**4 –** Satisfied.

**3 –** Neutral.

**2 –** Dissatisfied.

**1 –** Very dissatisfied.

## B.2. Hyperparameter Settings

In the SFT stage, we fine-tune the ECG-R1 for 1 epoch using full fine-tuning with a learning rate of 2e-5. Training is performed with a per-device batch size of 4 and gradient accumulation of 2 steps in 8 NVIDIA A100 GPUs. The ECG encoder and image encoder remain frozen. For IMD, we set the interleave probability $p_s$ to 0.1 and the modality dropout $p_d$ probability to 0.5. In the RL stage, we only train the LLM component while freezing the ECG encoder, image encoder, and their corresponding projectors. Training is performed for 1 epoch with a learning rate of 1e-6. We employ the DAPO algorithm with epsilon high set to 0.30 and epsilon low set to 0.2, and EDER weight $\lambda$ set to 1.0. Generation uses a temperature of 1.0 with dynamic sampling enabled. The training uses a per-device batch size of 4 and gradient accumulation of 2 steps. For IMD, we set the interleave probability $p_s$ to 0.1 and the modality dropout $p_d$ probability to 0.5.

## B.3. Baseline Models

We compare against four representative groups of baselines: (i) general-purpose proprietary MLLMs, including Gemini-3-Pro and GPT-5.1-Instant; (ii) general-purpose open-source MLLMs, including MiMo-VL-7B-SFT (Li et al., 2025c), GLM-4.1V-9B-Base (Team et al., 2026), Qwen3-VL-8B-Instruct (Bai et al., 2025), InternVL3-8B-Instruct (Zhu et al., 2025), and MiniCPM-V-4.5 (Yu et al., 2025b); (iii) medical MLLMs, including MedVLM-R1 (Pan et al., 2025), Chiron-o1-8B (Sun et al., 2026), QoQ-Med-VL-7B (Dai et al., 2025), MedGemma-4B/27B (Sellergren et al., 2025), MedGemma-27B (Sellergren et al., 2025), and HuatuoGPT-Vision-7B (Chen et al., 2024a); and (iv) ECG-specialized MLLMs, including ECG-Chat (Zhao et al., 2025a), PULSE (Liu et al., 2024c), and GEM (Lan et al., 2025), enabling a systematic comparison across general-purpose capability, open-source ecosystems, medical-domain adaptation, and ECG-specific specialization.

# C. Extended Experiment Results

## C.1. Dataset Comparison

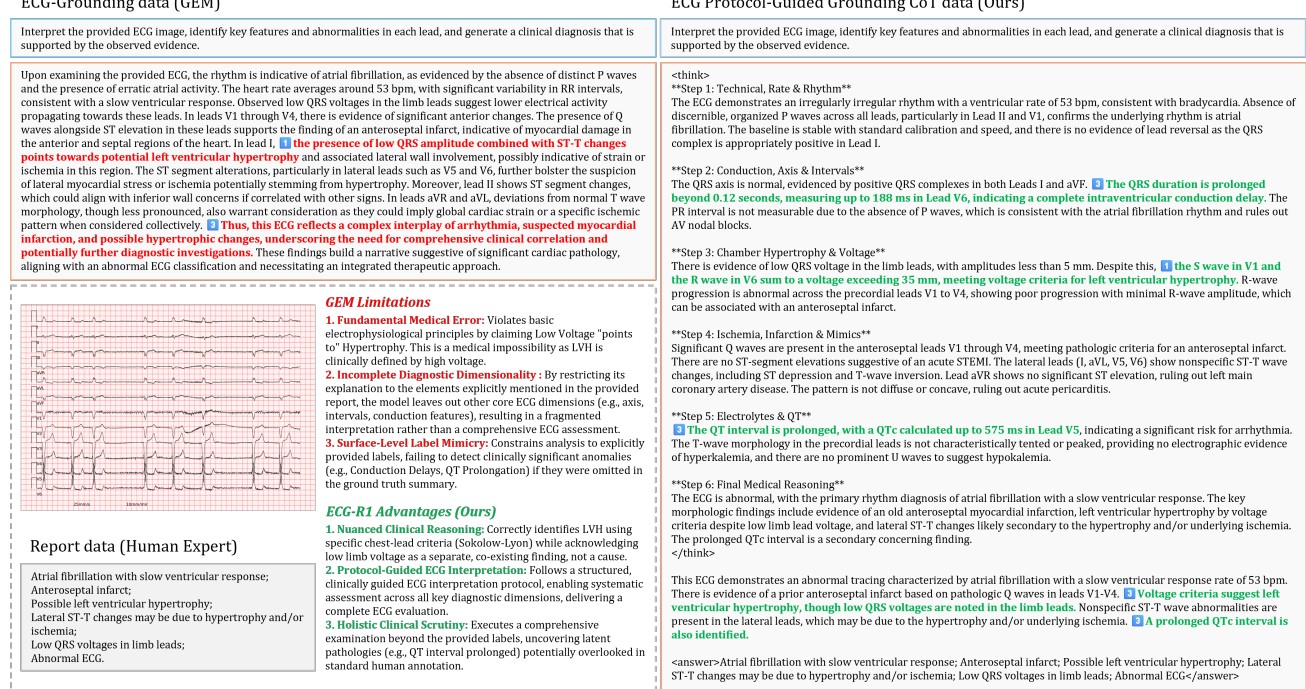

*Figure 5.* Qualitative Comparison of ECG-Grounding and our ECG Protocol-Guided Grounding CoT.

Figure 5 presents a qualitative comparison between two corpus generation paradigms, showing that ECG-Grounding data constructed under the GEM paradigm exhibits three systematic limitations: (1) Fundamental medical errors, where interpretations may violate basic electrocardiographic principles (e.g., conflating limb-lead low voltage with voltage-based LVH criteria); (2) Incomplete diagnostic dimensionality, as reasoning constrained to report-explicit elements often omits key dimensions such as intervals/conduction, axis assessment, and ischemia–infarction differentiation, resulting in fragmented assessments; and (3) Surface-level label imitation, where dependence on provided labels discourages identifying clinically relevant but unannotated findings (e.g., QTc prolongation or intraventricular conduction delay). In contrast, our ECG Protocol-Guided Grounding CoT data injects a standardized, verifiable interpretation protocol that tightly couples evidence extraction with stepwise reasoning, enabling more nuanced clinical inference, structured coverage across diagnostic dimensions, and more comprehensive scrutiny of latent high-risk abnormalities, thereby improving reliability and completeness.

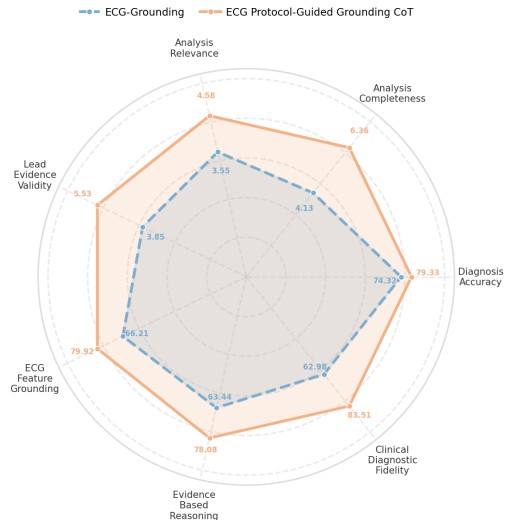

*Figure 6.* Quantitative Comparison of ECG-Grounding and our ECG Protocol-Guided Grounding CoT.

Figure 6 reports the quantitative results of ECG-R1 after SFT training on two datasets. Notably, the two datasets have the same number of training instances and identical ECG samples. The model architecture and hyperparameters are held constant, and no RL training is applied. Our ECG Protocol-Guided Grounding CoT consistently outperforms ECG-Grounding across

*Table 4.* Grounded ECG Interpretation Results Independently Evaluated by GLM-5.

| Metric | Diagnosis Accuracy | Analysis Completeness | Analysis Relevance | Lead Evidence Validity | ECG Feature Grounding | Evidence Based Reasoning | Clinical Diagnostic Fidelity |
|---|---|---|---|---|---|---|---|
| *Proprietary MLLMs* | | | | | | | |
| Gemini-3-Pro | 14.47 | 1.85 | 0.92 | 0.72 | 38.84 | 22.18 | 39.07 |
| GPT-5.1-Instant | 31.75 | 2.17 | 1.75 | 1.64 | 54.54 | 37.79 | 54.19 |
| *Open-source MLLMs* | | | | | | | |
| MiMo-VL-7B-SFT | 13.98 | 1.16 | 0.55 | 0.34 | 41.84 | 17.98 | 44.51 |
| GLM-4.1V-9B-Base | 15.27 | 0.91 | 0.48 | 0.40 | 31.19 | 16.89 | 28.68 |
| Qwen3-VL-8B-Instruct | 20.24 | 1.62 | 0.77 | 0.36 | 39.14 | 23.02 | 38.19 |
| InternVL3-8B-Instruct | 25.86 | 1.25 | 0.90 | 0.34 | 31.28 | 19.39 | 28.29 |
| MiniCPM-V-4.5 | 29.24 | 1.90 | 1.36 | 0.70 | 43.23 | 27.84 | 44.94 |
| *Medical MLLMs* | | | | | | | |
| MedVLM-R1 | 27.59 | 0.67 | 0.25 | 0.05 | 21.55 | 12.02 | 13.15 |
| Chiron-o1-8B | 23.97 | 2.14 | 1.28 | 0.74 | 38.64 | 21.98 | 32.01 |
| QoQ-Med-VL-7B | 33.19 | 2.19 | 1.79 | 0.66 | 46.96 | 28.97 | 46.49 |
| MedGemma-4B | 35.66 | 1.23 | 0.81 | 0.09 | 36.82 | 24.41 | 33.84 |
| MedGemma-27B | 29.23 | 1.92 | 1.38 | 1.02 | 48.07 | 30.48 | 49.22 |
| HuatuoGPT-Vision-7B | 33.81 | 2.82 | 1.89 | 0.31 | 44.85 | 31.52 | 38.21 |
| *ECG-specialized MLLMs* | | | | | | | |
| PULSE | 70.66 | 1.93 | 2.78 | 0.67 | 49.44 | 45.91 | 47.15 |
| GEM | 78.99 | 4.05 | 4.60 | **4.87** | 74.23 | 69.17 | 71.25 |
| ECG-R1 (SFT) | 83.07 | 3.99 | 4.74 | 4.53 | 86.74 | **81.06** | 88.58 |
| ECG-R1 (RL) | **83.29** | **4.18** | **4.87** | 4.66 | **86.96** | 81.02 | **88.90** |

all seven metrics. It improves diagnosis accuracy from 74.32 to 79.33 (+5.01) and yields substantially larger gains on process-oriented dimensions related to interpretability and traceability, with the largest increase in clinical diagnostic fidelity from 62.98 to 83.51 (+20.53), alongside improvements in evidence-based reasoning from 63.44 to 78.08 (+14.64) and ECG feature grounding from 66.21 to 79.92 (+13.71). Consistent improvements are also observed in analysis completeness from 4.13 to 6.36 (+2.23), analysis relevance from 3.55 to 4.58 (+1.03), and lead evidence validity from 3.85 to 5.53 (+1.68). The gains are driven not only by higher diagnosis accuracy but also by protocolized CoT supervision that strengthens process constraints and evidence alignment during SFT, leading to more verifiable intermediate evidence and clinically faithful reasoning without sacrificing final correctness.

## C.2. Grounded ECG Interpretation Results Independently Evaluated by GLM-5

To facilitate future research and reproducible benchmarking, we additionally conduct an independent evaluation with GLM-5, a recent and accessible open-source LLM evaluator, under the same rubric. We select representative models from each category and report the results in Table 4. Since LLM-based evaluators may encode model-specific biases in their generated judgments, the absolute scores produced by GLM-5 may differ from those in the main evaluation. Across evaluator choices, the overall trend remains consistent: ECG-specialized MLLMs outperform general-purpose and medical MLLMs on grounded ECG interpretation, and ECG-R1 achieves the strongest overall performance across most dimensions. This complementary evaluation further supports the robustness of our findings and provides an accessible reference for future comparisons.

## C.3. ECG-Bench Results

We further evaluate ECG-R1 on conventional ECG abnormality detection using ECG-Bench (Liu et al., 2024c), as shown in Table 5. We use macro AUC, macro F1, and hamming loss (HL) for multi-label datasets, and accuracy for others. Across all five datasets, proprietary MLLMs exhibit near-random performance, indicating that generic multimodal capabilities are insufficient for accurate ECG abnormality detection. We compare against ECG-specialized methods, where ST-MEM is

*Table 5.* ECG-Bench Abnormality Detection Results.

| Datasets | PTB-XL Super | | | CODE-15% | | | CPSC 2018 | | | CSN | G12EC |
|---|---|---|---|---|---|---|---|---|---|---|---|
| Metric | AUC | F1 | HL | AUC | F1 | HL | AUC | F1 | HL | ACC | ACC |
| Random | 50.3 | 33.2 | 50.1 | 48.8 | 15.0 | 32.1 | 51.2 | 15.1 | 28.8 | 11.6 | 12.1 |
| *Proprietary MLLMs* | | | | | | | | | | | |
| GPT-4o | 55.6 | 28.3 | 26.2 | 59.9 | 24.9 | 15.7 | 50.9 | 10.6 | 18.2 | 57.5 | 49.2 |
| GPT-4o-mini | 52.0 | 20.4 | 31.7 | 57.5 | 22.0 | 15.1 | 49.2 | 11.0 | 25.5 | 32.1 | 33.2 |
| Gemini-1.5-Pro | 50.7 | 15.3 | 27.9 | 56.7 | 20.0 | 15.9 | 50.1 | 7.4 | 20.5 | 50.5 | 36.0 |
| Claude-3.5-Sonnet | 54.0 | 27.5 | 29.6 | 58.3 | 20.3 | 17.8 | 52.8 | 11.5 | 18.9 | 51.5 | 51.4 |
| *ECG-specialized Methods* | | | | | | | | | | | |
| METS | - | 65.7 | - | - | - | - | - | - | - | N/A | N/A |
| MERL | 74.2 | - | - | - | - | - | 82.8 | - | - | N/A | N/A |
| ST-MEM | 71.4 | - | - | - | - | - | 70.4 | - | - | N/A | N/A |
| PULSE | 82.4 | 74.8 | **11.0** | 90.7 | 85.4 | 5.0 | 76.9 | 57.6 | 8.6 | 85.2 | 78.2 |
| GEM | **83.4** | **75.8** | 11.0 | **91.5** | 86.4 | 4.7 | 79.1 | **61.1** | **8.1** | 86.2 | 80.5 |
| ECG-R1(Ours) | 81.7 | 73.7 | 11.4 | 91.4 | **86.7** | **4.6** | 74.9 | 51.2 | 9.8 | **90.4** | **84.5** |

a self-supervised ECG model, METS (Li et al., 2024) and MERL (Liu et al., 2024b) are CLIP-like multimodal models, PULSE (Liu et al., 2024c), and GEM (Lan et al., 2025) are ECG-oriented MLLMs. Although ECG-R1 is not explicitly optimized for abnormality detection, it remains competitive with top-performing approaches, achieving new state of the art on CSN and G12EC, while showing only marginal gaps to the best methods on the remaining datasets.

## C.4. Ablation Study on Interleaved Modality Dropout

*Table 6.* Effect of Interleaved Modality Dropout on Robustness.

| Metric | Diagnosis Accuracy | | Peak Performance Trade-off | Robustness Recovery Gain | Efficiency Ratio |
|---|---|---|---|---|---|
| | Omni Modalities (Time-Series + Image) | Modality Missing (Only Time-Series) | | | |
| w/o IMD | 81.99 | 36.77 | - | - | - |
| w/ IMD | 80.29 | 77.91 | -1.7 | 41.14 | 24.2x |

*Table 7.* Effect of Interleaved Modality Dropout on Consistency.

| Metric | BLEU-4 | ROUGE-L | SBERT-Score |
|---|---|---|---|
| w/o IMD | 0.54 | 0.59 | 0.92 |
| w/ IMD | 0.69 | 0.73 | 0.97 |

Table 6 studies how enabling Interleaved Modality Dropout (IMD) during training affects robustness. Omni modalities refers to evaluation with both time-series and image inputs, while modality missing refers to evaluation with the image modality completely absent and only time-series signals provided. Without IMD, the model achieves slightly higher peak accuracy under omni-modality inputs, but its performance collapses under modality missing, dropping to 36.77. In contrast, training with IMD maintains strong performance when the image modality is absent, reaching 77.91 in modality missing. This corresponds to a robustness recovery gain of 41.14 with only a modest peak-performance trade-off of 1.7. Overall, IMD substantially improves robustness under modality missing conditions while incurring only a minor reduction in omni modality accuracy, resulting in an efficiency ratio of 24.2x, defined as the ratio between robustness recovery gain and peak-performance trade-off.

Table 7 reports the effect of Interleaved Modality Dropout on cross-modality consistency of model interpretations. For each sample in the test set, we generate two interpretations, one conditioned on the image modality only and the other conditioned on the time-series modality only, and compute BLEU-4, ROUGE-L, and SBERT-Score to measure the agreement between

the two outputs. We then average these scores over all test samples to obtain the reported results. Without IMD, the two single-modality interpretations show limited alignment, indicating substantial variability when the available modality changes. Enabling IMD yields higher cross-modality consistency, achieving BLEU-4 of 0.69, ROUGE-L of 0.73, and SBERT-Score of 0.97 averaged over the entire test set, demonstrating that IMD promotes more modality-invariant and semantically consistent interpretations.

Overall, IMD substantially improves both robustness and cross-modality consistency of the model outputs. The empirical gains are consistent with our theoretical analysis, suggesting that IMD is a sound training strategy for improving reliability under modality missing conditions.

### C.5. Ablation Study on ECG Diagnostic Evidence Rewards

*Table 8.* Effect of EDER on Grounded ECG Interpretation Results.

| Metric | Diagnosis Accuracy | Analysis Completeness | Analysis Relevance | Lead Evidence Validity | ECG Feature Grounding | Evidence Based Reasoning | Clinical Diagnostic Fidelity |
|---|---|---|---|---|---|---|---|
| w/o EDER | 79.96 | 5.94 | 4.30 | 4.72 | 78.53 | 78.14 | 83.53 |
| w/ EDER | 80.29 | 6.51 | 4.74 | 5.81 | 80.57 | 79.08 | 84.20 |

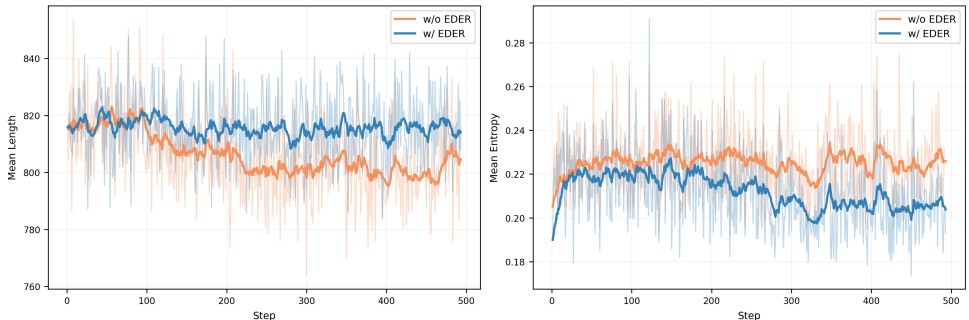

*Figure 7.* Effect of EDER on Mean Output Length and Entropy during RL Training.

Table 8 summarizes the effect of incorporating ECG Diagnostic Evidence Rewards (EDER) during RL training. EDER improves both terminal diagnosis accuracy and the quality of the generated interpretations, consistently boosting analysis completeness and relevance, lead evidence validity, ECG feature grounding, evidence-based reasoning, and clinical diagnostic fidelity. We provide a deeper analysis of the above results. Figure 7 shows that incorporating EDER substantially alters the MLLM's generation behavior during RL training. Without EDER, the mean rollout length progressively contracts, indicating that when optimization is dominated by terminal diagnostic correctness, the policy can degenerate into a "shorter answer yields comparable reward" strategy, thereby omitting key diagnostic evidence statements. With EDER, the rollout length remains more stable, suggesting that process-level rewards impose sustained constraints on evidence-driven interpretation, mitigating length collapse and encouraging the preservation of salient evidence statements. Regarding uncertainty, with EDER the output entropy is generally lower and exhibits smaller fluctuations, implying that EDER provides clearer learning signals that concentrate the generation distribution and promote more consistent decisions, leading to a more stable training trajectory. Conversely, without EDER, entropy remains higher and fluctuates more, suggesting that in the absence of evidence-level shaping, policy updates are driven primarily by sparse end-point feedback, resulting in less consistent generation dynamics and noisier training trajectories. Overall, EDER improves both the completeness of the reasoning process and the stability of generation, better aligning the learned policy with clinical requirements for interpretable and auditable ECG interpretation.

## C.6. Numerical Analysis of the DiagnosisAccuracy Metric Evaluated by GPT-4o and DeepSeek v3.1 Terminus

Our observations and analyses indicate that the scoring mechanism of GPT-4o is predisposed toward a semantic-relevance-based assessment. It is frequently confounded by secondary information generated by the model—such as accurate heart rate or rhythm descriptions—leading to an excessive tolerance for fatal clinical omissions (false negatives) and logically inconsistent over-diagnoses (false positives). This "semantic-priority" logic entails significant risk in clinical scenarios. Conversely, the evaluation logic of DeepSeek v3.1 Terminus manifests as a rigorous clinical audit, prioritizing the accuracy of core diagnostic conclusions.

Furthermore, GPT-4o's tendency to aggregate multiple diagnostic points into a single score results in a minimal denominator ($N$), leading to significant metric inflation. Under a binary calculation logic where any score greater than 0 is categorized as correct, even marginal partial credits are amplified into a 100% accuracy rate for the given sample. This masks the model's performance bottlenecks in complex cases. In contrast, DeepSeek v3.1 Terminus mandates an exhaustive decomposition of diagnostic claims, substantially increasing the evaluation denominator, thereby enhancing both evaluative transparency and statistical discriminability.

To further illustrate these systematic discrepancies in evaluation logic and their impact on quantitative outcomes, we present three representative case studies. These cases exemplify the divergent behaviors of GPT-4o and DeepSeek v3.1 Terminus in handling clinical omissions, evaluation granularity, and diagnostic hallucinations:

### Sample 45108524

**GPT-4o Evaluation Results:**
- The generated interpretation identifies a normal sinus rhythm correctly but lacks the specific mention of potential ischemic or infarct patterns seen in the ground truth. It mostly aligns with the identified rhythm but misses key findings suggestive of ischemia. **(1.0)**

**DeepSeek v3.1 Terminus Evaluation Results:**
- The generated interpretation concludes a 'normal ECG' with 'no evidence of ischemia, infarction, or conduction abnormalities.' This is incorrect and non-specific. The ground truth identifies key, specific diagnoses supported by ECG findings. **(0.0)**

### Sample 44425880

**GPT-4o Evaluation Results:**
- The generated interpretation correctly identifies sinus bradycardia (2.0) and an inferior infarct (2.0), supported by the findings of prolonged PR interval and QRS amplitude abnormalities. However, it lacks specificity and clear support for the diagnosis of lateral ischemia. **(4.0)**

**DeepSeek v3.1 Terminus Evaluation Results:**
- Correctly identified sinus bradycardia with supporting heart rate (55 bpm). **(2.0)**
- Correctly identified borderline first-degree A-V block (214 ms). **(2.0)**
- Correctly identified inferior infarct with supporting findings (abnormal Q waves). **(2.0)**
- Identified lateral ischemia but lacked specificity and over-interpreted ST changes in Lead I/aVL as definitive. **(1.0)**
- Incorrectly suggested anterior ischemia (V1-V4) without ground truth support. **(0.0)**
- Incorrectly suggested left main disease from aVR upslope, which was not mentioned in ground truth. **(0.0)**

### Sample 43556732

**GPT-4o Evaluation Results:**
- The generated interpretation correctly identifies sinus rhythm with PACs and a first-degree A-V block. However, the left ventricular hypertrophy and inferior infarct suggestions are less clear. **(2.0)**
- The interpretation mentioned possible left atrial enlargement and lateral ischemia, which are noted inconsistently with ECG features. **(1.0)**

**DeepSeek v3.1 Terminus Evaluation Results:**
- Correctly identified sinus rhythm with PACs and first-degree A-V block, with specific ECG support (irregular RR intervals, PR interval of 280 ms). **(2.0)**
- Diagnosis of 'inferior infarct' is incorrect and over-interprets findings. **(0.0)**
- Diagnosis of 'lateral ischemia' is vague and not supported by ground truth. **(0.0)**
- Diagnosis of 'left main coronary artery disease' based on aVR is incorrect. **(0.0)**

In medical research and clinical decision support systems, lenience in assessment tools often equates to the condoning of medical risks. Therefore, utilizing DeepSeek v3.1 Terminus as the evaluative model provides a benchmark with superior clinical reference value and statistical power for medical LLMs. This explains why, despite lower absolute values in the $Diagnosis Accuracy$ metric, the evaluative quality provided by DeepSeek v3.1 Terminus is far superior to that of GPT-4o.

### C.7. Limitations and Future Work

While ECG-R1 demonstrates strong performance in evidence-grounded ECG interpretation, several directions remain for future improvement. Although EDER improves performance both within and outside the top-500 report patterns, the RL data are concentrated on frequent reports. Rare but clinically important conditions therefore require further targeted data curation and reward optimization. In addition, QT/QTc interval estimation remains challenging due to uneven numerical supervision in the training corpus. Future work will incorporate explicit measurement-aware supervision, such as auxiliary numerical regression objectives and calibrated uncertainty estimation, to improve the reliability of interval-level ECG measurements.

## D. Complete Proofs for Section 2.4

**Notation.** Let $x = (x^{\text{text}}, x^I, x^T)$ be the multimodal input and $y$ the output text sequence. For a transformation $\tau$ (dropping a modality or swapping modality-token blocks), define the environment-specific observation

$$z_\tau \triangleq \tau(x),$$

and the induced model distribution

$$P_\theta^\tau(\cdot \mid x) \triangleq P_\theta(\cdot \mid z_\tau) = P_\theta(\cdot \mid \tau(x)).$$

We train with negative log-likelihood (NLL)

$$\ell_\theta(\tau(x), y) \triangleq -\log P_\theta(y \mid \tau(x)).$$

The population risk in environment $\tau$ is

$$R_\tau(\theta) \triangleq \mathbb{E}_{(x,y) \sim \mathcal{D}}\big[\ell_\theta(\tau(x), y)\big] = \mathbb{E}_{x \sim \mathcal{D}}\mathbb{E}_{y \sim P^\star(\cdot \mid \tau(x))}\big[-\log P_\theta(y \mid \tau(x))\big], \tag{7}$$

where $P^\star(\cdot \mid \tau(x))$ denotes the ground-truth conditional distribution of $y$ given the observation $z_\tau = \tau(x)$. For IMD, we consider the finite set

$$\mathcal{T}_{\text{test}} = \{\tau_I, \tau_T, \tau_{IT}, \tau_{TI}\}.$$

IMD samples $\tau \sim q$ over $\mathcal{T}_{\text{test}}$ via two independent trials: (i) a modality-drop trial with probability $p_d$; and (ii) conditioned on retaining both modalities, a token-order swap trial with probability $p_s$. Concretely,

$$q(\tau_I) = q(\tau_T) = \frac{p_d}{2}, \qquad q(\tau_{TI}) = (1 - p_d)(1 - p_s), \qquad q(\tau_{IT}) = (1 - p_d)p_s,$$

where $\tau_I, \tau_T$ drop one modality and $\tau_{TI}, \tau_{IT}$ correspond to the canonical and swapped token-block orders, respectively. We choose $p_d \in (0, 1)$ and $p_s \in (0, 1)$ so that $q(\tau) > 0$ for all $\tau \in \mathcal{T}_{\text{test}}$. Therefore, Assumption 2.1 holds with

$$\alpha = \min\left\{\frac{p_d}{2}, (1 - p_d)(1 - p_s), (1 - p_d)p_s\right\}.$$

The mixture risk under sampling distribution $q$ is

$$R_q(\theta) \triangleq \mathbb{E}_{\tau \sim q}\big[R_\tau(\theta)\big]. \tag{8}$$

We also define the mixture Bayes risk $\bar{R}_q^\star \triangleq \mathbb{E}_{\tau \sim q}[R_\tau^\star]$.

**Bayes-optimal risks and excess risks.** Define the Bayes-optimal (unconstrained) risk for each environment:

$$R_\tau^\star \triangleq \inf_{Q(\cdot \mid z)} \mathbb{E}_{x \sim \mathcal{D}}\mathbb{E}_{y \sim P^\star(\cdot \mid \tau(x))}\big[-\log Q(y \mid z)\big]\Big|_{z = \tau(x)} = \mathbb{E}_{x \sim \mathcal{D}} H\big(P^\star(\cdot \mid \tau(x))\big), \tag{9}$$

where the infimum is over all conditional distributions $Q(\cdot \mid z)$ defined on the environment observation $z = \tau(x)$.

We define the excess risk of model $\theta$ in environment $\tau$ as

$$\varepsilon_\tau(\theta) \triangleq R_\tau(\theta) - R_\tau^\star. \tag{10}$$

For the single-modality environments, we also define the intrinsic view gap

$$\Delta_{\text{view}} \triangleq \mathbb{E}_{x \sim \mathcal{D}}\Big[\text{TV}\big(P_{\tau_I}^\star(\cdot \mid x), P_{\tau_T}^\star(\cdot \mid x)\big)\Big], \tag{11}$$

which can be non-zero due to information disparity between modalities.

## D.1. Auxiliary Lemmas

**Lemma D.1** (Cross-entropy decomposition). *For any conditional distributions $P^\star(\cdot \mid x)$ and $Q(\cdot \mid x)$,*

$$\mathbb{E}_{y \sim P^\star(\cdot \mid x)}[-\log Q(y \mid x)] = H(P^\star(\cdot \mid x)) + D_{\mathrm{KL}}\big(P^\star(\cdot \mid x) \,\|\, Q(\cdot \mid x)\big).$$

*Consequently, for each environment $\tau$,*

$$R_\tau(\theta) = R_\tau^\star + \mathbb{E}_{x \sim \mathcal{D}}\Big[D_{\mathrm{KL}}\big(P_\tau^\star(\cdot \mid x) \,\|\, P_\theta^\tau(\cdot \mid x)\big)\Big], \quad \text{and hence} \quad \varepsilon_\tau(\theta) = \mathbb{E}_x D_{\mathrm{KL}}\big(P_\tau^\star \| P_\theta^\tau\big). \tag{12}$$

*Proof.* The identity is standard: cross-entropy equals entropy plus KL divergence. Taking expectation over $x \sim \mathcal{D}$ and substituting $Q = P_\theta^\tau$ yields the first equality. By definition of $R_\tau^\star$ in (9), $R_\tau^\star = \mathbb{E}_x H(P_\tau^\star(\cdot \mid x))$. Subtracting completes (12). $\square$

**Lemma D.2** (Pinsker's inequality). *For any distributions $P, Q$,*

$$\mathrm{TV}(P, Q) \le \sqrt{\tfrac{1}{2} D_{\mathrm{KL}}(P\|Q)}.$$

**Lemma D.3** (Jensen for square root). *For any nonnegative random variable $Z$, $\mathbb{E}[\sqrt{Z}] \le \sqrt{\mathbb{E}[Z]}$.*

## D.2. Proof of Robustness Theorem

**Theorem (Robustness under IMD).** Assume **Coverage**: there exists $\alpha > 0$ such that $q(\tau) \ge \alpha$ for all $\tau \in \mathcal{T}_{\text{test}}$. Define

$$R_{\max}(\theta) \triangleq \max_{\tau \in \mathcal{T}_{\text{test}}} R_\tau(\theta).$$

Then

$$R_{\max}(\theta) \le \alpha^{-1} R_q(\theta). \tag{13}$$

**Proof.** Let $\tau^\star \in \arg\max_{\tau \in \mathcal{T}_{\text{test}}} R_\tau(\theta)$ so that $R_{\max}(\theta) = R_{\tau^\star}(\theta)$. By the definition of mixture risk (8),

$$R_q(\theta) = \mathbb{E}_{\tau \sim q}[R_\tau(\theta)] \ge q(\tau^\star)\, R_{\tau^\star}(\theta) \ge \alpha\, R_{\max}(\theta),$$

which implies (13). $\square$

## D.3. Proof of Consistency and Swap-Invariance Theorem

**Definitions (consistency metrics).** For the single-modality views, define

$$\mathcal{F}(\theta) \triangleq \mathbb{E}_{x \sim \mathcal{D}}\, \mathrm{TV}\big(P_\theta^{\tau_I}(\cdot \mid x),\, P_\theta^{\tau_T}(\cdot \mid x)\big).$$

For interleaving (block-swap) invariance, define

$$\mathcal{F}_{\text{swap}}(\theta) \triangleq \mathbb{E}_{x \sim \mathcal{D}}\, \mathrm{TV}\big(P_\theta^{\tau_{IT}}(\cdot \mid x),\, P_\theta^{\tau_{TI}}(\cdot \mid x)\big).$$

Optionally, one may also define an intrinsic swap gap $\Delta_{\text{swap}} \triangleq \mathbb{E}_x \mathrm{TV}(P_{\tau_{IT}}^\star(\cdot \mid x), P_{\tau_{TI}}^\star(\cdot \mid x))$, which is typically 0 when only the token block order changes the input representation.

**Theorem (Consistency via excess risk).** For any $\theta$,

$$\mathcal{F}(\theta) \le \Delta_{\text{view}} + \sqrt{\varepsilon_{\tau_I}(\theta)/2} + \sqrt{\varepsilon_{\tau_T}(\theta)/2}, \tag{14}$$

$$\mathcal{F}_{\text{swap}}(\theta) \le \Delta_{\text{swap}} + \sqrt{\varepsilon_{\tau_{IT}}(\theta)/2} + \sqrt{\varepsilon_{\tau_{TI}}(\theta)/2}. \tag{15}$$

Moreover, letting $\bar{R}_q^\star \triangleq \mathbb{E}_{\tau \sim q}[R_\tau^\star]$, for any $\tau$,

$$R_q(\theta) - \bar{R}_q^\star \ge q(\tau)\, \varepsilon_\tau(\theta), \quad \text{hence under Coverage } (q(\tau) \ge \alpha) \text{ we have } R_q(\theta) - \bar{R}_q^\star \ge \alpha\, \varepsilon_\tau(\theta). \tag{16}$$

**Proof of** (14). Fix any $x$ and abbreviate $P_I^\theta(\cdot) = P_\theta^{\tau_I}(\cdot \mid x)$, $P_T^\theta(\cdot) = P_\theta^{\tau_T}(\cdot \mid x)$, $P_I^\star(\cdot) = P_{\tau_I}^\star(\cdot \mid x)$, $P_T^\star(\cdot) = P_{\tau_T}^\star(\cdot \mid x)$. By triangle inequality for total variation,

$$\mathrm{TV}(P_I^\theta, P_T^\theta) \le \mathrm{TV}(P_I^\theta, P_I^\star) + \mathrm{TV}(P_I^\star, P_T^\star) + \mathrm{TV}(P_T^\star, P_T^\theta). \tag{17}$$

Taking expectation over $x \sim \mathcal{D}$ gives

$$\mathcal{F}(\theta) \le \Delta_{\text{view}} + \mathbb{E}_x \mathrm{TV}(P_\theta^{\tau_I}(\cdot \mid x), P_{\tau_I}^\star(\cdot \mid x)) + \mathbb{E}_x \mathrm{TV}(P_{\tau_T}^\star(\cdot \mid x), P_\theta^{\tau_T}(\cdot \mid x)). \tag{18}$$

Apply Pinsker's inequality (Lemma D.2) to the two TV terms:

$$\mathrm{TV}(P_I^\star, P_I^\theta) \le \sqrt{\tfrac{1}{2} D_{\mathrm{KL}}(P_I^\star \| P_I^\theta)}, \qquad \mathrm{TV}(P_T^\star, P_T^\theta) \le \sqrt{\tfrac{1}{2} D_{\mathrm{KL}}(P_T^\star \| P_T^\theta)}.$$

Taking expectation and applying Jensen (Lemma D.3) yields

$$\mathbb{E}_x \mathrm{TV}(P_I^\star, P_I^\theta) \le \sqrt{\tfrac{1}{2} \mathbb{E}_x D_{\mathrm{KL}}(P_{\tau_I}^\star(\cdot \mid x) \| P_\theta^{\tau_I}(\cdot \mid x))}, \tag{19}$$

$$\mathbb{E}_x \mathrm{TV}(P_T^\star, P_T^\theta) \le \sqrt{\tfrac{1}{2} \mathbb{E}_x D_{\mathrm{KL}}(P_{\tau_T}^\star(\cdot \mid x) \| P_\theta^{\tau_T}(\cdot \mid x))}. \tag{20}$$

Finally, by Lemma D.1 (Eq. (12)),

$$\mathbb{E}_x D_{\mathrm{KL}}(P_{\tau_I}^\star \| P_\theta^{\tau_I}) = \varepsilon_{\tau_I}(\theta), \qquad \mathbb{E}_x D_{\mathrm{KL}}(P_{\tau_T}^\star \| P_\theta^{\tau_T}) = \varepsilon_{\tau_T}(\theta).$$

Substituting into (18) proves (14). □

**Proof of** (15). The proof is identical to the view case, replacing $(\tau_I, \tau_T)$ with $(\tau_{IT}, \tau_{TI})$ and $\Delta_{\text{view}}$ with $\Delta_{\text{swap}}$. □

**Proof of mixture dominance** (16). By definitions,

$$R_q(\theta) - \bar{R}_q^\star = \mathbb{E}_{\tau \sim q}[R_\tau(\theta)] - \mathbb{E}_{\tau \sim q}[R_\tau^\star] = \mathbb{E}_{\tau \sim q}\big[R_\tau(\theta) - R_\tau^\star\big] = \mathbb{E}_{\tau \sim q}\big[\varepsilon_\tau(\theta)\big].$$

Since $\varepsilon_\tau(\theta) \ge 0$, for any fixed $\tau_0$,

$$\mathbb{E}_{\tau \sim q}[\varepsilon_\tau(\theta)] \ge q(\tau_0)\, \varepsilon_{\tau_0}(\theta),$$

which gives the first inequality in (16). If additionally $q(\tau_0) \ge \alpha$, then $R_q(\theta) - \bar{R}_q^\star \ge \alpha\, \varepsilon_{\tau_0}(\theta)$. □

**D.4. Remarks on Using $\inf_\theta R_\tau(\theta)$**

**Remark.** In the main text, one may write $R_\tau^\star = \inf_\theta R_\tau(\theta)$ for notational simplicity. The identities in Lemma D.1 and Eq. (12) hold exactly for the Bayes-optimal risk defined in (9). If the model family is expressive enough to realize $P_\tau^\star(\cdot \mid x)$ (for each $\tau \in \mathcal{T}_{\text{test}}$), then $\inf_\theta R_\tau(\theta) = R_\tau^\star$ and all bounds remain unchanged. Otherwise, one can interpret $\varepsilon_\tau(\theta)$ as excess risk over the Bayes-optimal predictor; replacing it by $R_\tau(\theta) - \inf_\theta R_\tau(\theta)$ introduces an additional approximation term that is standard in statistical learning.

# E. Case Study

## E.1. Inference Results

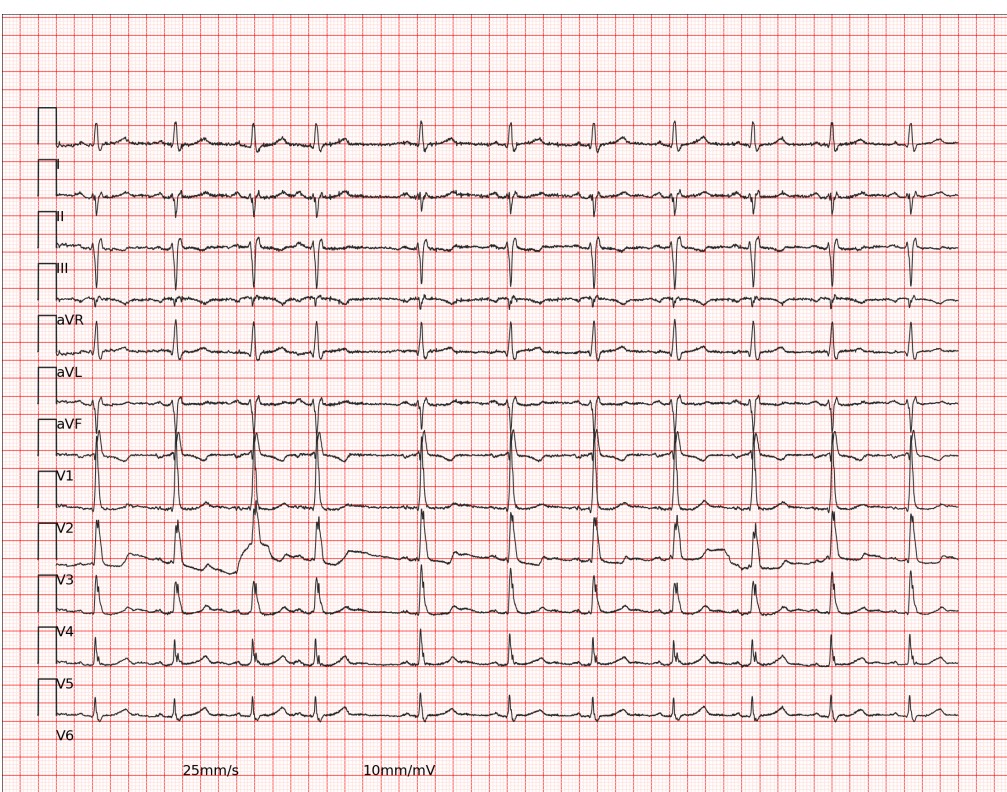

*Figure 8.* Report: Sinus rhythm with PACs; Left axis deviation; RBBB with left anterior fascicular block; Abnormal ECG.

### GEM Interpretation

Upon analyzing the provided ECG image and computed measurements, several key features and abnormalities are evident. The rhythm is sinus with occasional premature atrial contractions (PACs), as indicated by irregular RR intervals, particularly the shorter intervals such as 238 ms and 241 ms. The heart rate is 68 bpm, which is within normal limits. The presence of left axis deviation is suggested by the negative QRS complex in lead aVF and positive in lead I. The QRS duration is prolonged, particularly in leads I and V1, with durations reaching up to 150 ms, indicating a right bundle branch block (RBBB). Additionally, the left anterior fascicular block is suggested by the left axis deviation and the presence of a small R wave in lead I.

Inferior infarction is indicated by abnormal Q waves and ST segment changes in leads II, III, and aVF. The QRS amplitude is notably reduced in these leads, and the ST segment shows a declination form, which is consistent with ischemic changes. The anterior leads, particularly V2 to V4, show some ST segment elevation, which could suggest an anterior myocardial infarction. However, the lack of significant Q waves in these leads makes it less definitive.

Overall, the ECG findings are consistent with a diagnosis of sinus rhythm with PACs, left axis deviation, RBBB with left anterior fascicular block, and evidence of an inferior infarct. The anterior ST elevation is suggestive of possible myocardial injury, but further clinical correlation and possibly additional testing would be necessary to confirm this. The ECG is abnormal, and these findings warrant further investigation and clinical management.

## ECG-R1 Interpretation

<think>

### Step 1: Technical, Rate & Rhythm

The ECG demonstrates good technical quality with a stable baseline, standard 25 mm/s speed, and 10 mm/mV calibration, with no evidence of lead reversal as P waves are upright in lead I. The rhythm is sinus with a rate of 66 bpm, confirmed by the presence of upright P waves in leads II, III, and aVF. The rhythm is occasionally irregular, with intermittent premature beats characterized by an early, abnormal P wave morphology and a compensatory pause, consistent with premature atrial contractions (PACs); the absence of an irregularly irregular rhythm without P waves rules out atrial fibrillation.

### Step 2: Conduction, Axis & Intervals

The frontal plane axis demonstrates left axis deviation, evidenced by a positive QRS complex in lead I and a negative QRS complex in lead II. The PR interval is within normal limits at approximately 160 ms, ruling out AV block. The QRS duration is prolonged at 120 ms, and an rSR' pattern is present in lead V1 with a wide, slurred S wave in leads I and V6, confirming a right bundle branch block (RBBB). The combination of left axis deviation with RBBB is indicative of a bifascicular block involving the right bundle and the left anterior fascicle.

### Step 3: Chamber Hypertrophy & Voltage

There is no evidence of left ventricular hypertrophy, as the S wave in V1 and R wave in V6 do not meet the Sokolow-Lyon voltage criteria. R wave progression across the precordial leads V1 to V4 is normal, with a transition zone around V3-V4, which rules out conditions like poor R wave progression or reversed progression suggestive of right ventricular hypertrophy or posterior infarction.

### Step 4: Ischemia, Infarction & Mimics

A systematic analysis of all lead groups reveals no acute ischemic changes. There is no significant ST-segment elevation or depression meeting threshold criteria in the inferior (II, III, aVF), lateral (I, aVL, V5, V6), or anterior (V1-V4) leads. The ST segments are predominantly isoelectric or show non-specific, minimal deviations. No pathologic Q waves are present in any territory. The T waves are appropriately discordant to the QRS complex in leads with RBBB (V1-V2), which is a normal finding and not indicative of ischemia.

### Step 5: Electrolytes & QT

The T wave morphology is normal without evidence of tenting suggestive of hyperkalemia or flattening with U waves suggestive of hypokalemia. The corrected QT interval (QTc) is within normal limits, calculated at approximately 440 ms, ruling out significant QT prolongation.

### Step 6: Final Medical Reasoning

The ECG findings are consistent with a sinus rhythm with occasional PACs, a bifascicular block pattern comprising right bundle branch block and left anterior fascicular block, and no evidence of acute ischemia, infarction, or electrolyte disturbance. The overall tracing is abnormal due to the conduction delays.

</think>

This ECG demonstrates sinus rhythm at 66 bpm with occasional premature atrial contractions. The conduction system shows a bifascicular block pattern, characterized by a right bundle branch block with a QRS duration of 120 ms and left axis deviation, consistent with concomitant left anterior fascicular block. There is no evidence of chamber hypertrophy, acute ischemic ST-T wave changes, pathologic Q waves, or electrolyte abnormalities. The corrected QT interval is normal. The tracing is abnormal due to the identified conduction delays.

<answer>Sinus rhythm with PAC(s); Left axis deviation; RBBB with left anterior fascicular block; Abnormal ECG</answer>

**Case Study: 48410813**

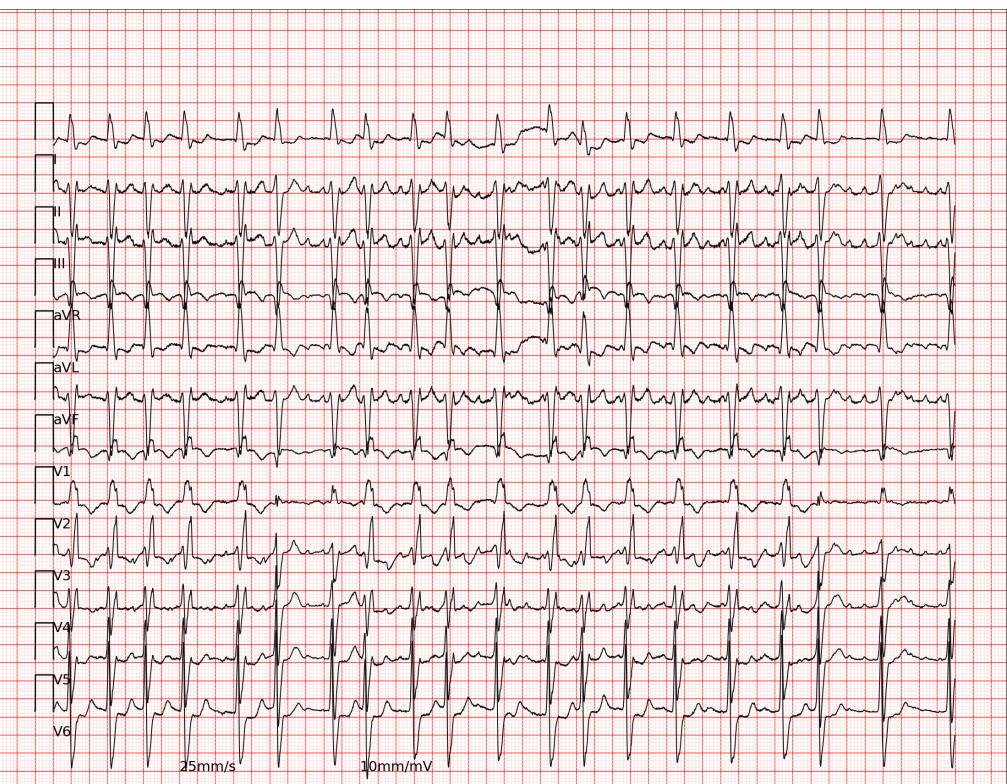

*Figure 9.* Report: Atrial fibrillation with rapid ventricular response with PVC(s); Left axis deviation; RBBB with left anterior fascicular block; Lateral ST-T changes may be due to myocardial ischemia; Abnormal ECG.

**GEM Interpretation**

Upon analyzing the provided ECG image and computed measurements, several key features and abnormalities are evident. The heart rate is elevated at 120 bpm, indicating sinus tachycardia. The presence of premature atrial contractions (PACs) is suggested by irregular RR intervals, particularly the shorter intervals such as 188 ms and 203 ms. The ECG also reveals a left axis deviation, which is supported by the negative QRS complex in lead aVF and positive in lead I. The QRS duration is prolonged in several leads, notably in lead I with durations up to 158 ms, indicating a right bundle branch block (RBBB) with left anterior fascicular block. This is further corroborated by the presence of a small R wave in lead V1 and a broad S wave in lead I.

Additionally, the ECG shows signs of left ventricular hypertrophy, as suggested by the increased QRS amplitude in leads V5 and V6, with values reaching up to 1.7 mV. The lateral ST-T changes, particularly in leads I, aVL, V5, and V6, are likely due to ventricular hypertrophy, as indicated by the ST segment declination and T wave inversion. These findings collectively suggest an abnormal ECG with multiple underlying cardiac conditions. The combination of sinus tachycardia, PACs, left axis deviation, RBBB with left anterior fascicular block, and left ventricular hypertrophy, along with lateral ST-T changes, provides a comprehensive picture of the patient's cardiac status. This analysis is consistent with the observed ECG features and computed measurements, supporting a diagnosis of an abnormal ECG with multiple cardiac abnormalities.

### ECG-R1 Interpretation

<think>

**Step 1: Technical, Rate & Rhythm**

The ECG demonstrates an irregularly irregular rhythm with a mean ventricular rate of 118 bpm, consistent with tachycardia. There is an absence of discernible, organized P waves across all leads, including Lead II and V1, which rules out sinus rhythm and confirms the diagnosis of atrial fibrillation. The baseline shows some undulation but is otherwise stable without significant artifact, and the calibration is standard at 25 mm/s and 10 mm/mV, with no evidence of limb lead reversal as P waves are not present to assess.

**Step 2: Conduction, Axis & Intervals**

The frontal plane axis demonstrates left axis deviation, evidenced by a positive QRS complex in Lead I and a negative QRS complex in Lead II. The QRS duration is prolonged at 120-130 ms, and an rSR' pattern is present in Lead V1 with a wide, slurred S wave in Leads I and V6, confirming a right bundle branch block (RBBB). The combination of left axis deviation with RBBB is indicative of a concomitant left anterior fascicular block. The PR interval is not measurable due to the underlying atrial fibrillation.

**Step 3: Chamber Hypertrophy & Voltage**

There is evidence of left ventricular hypertrophy based on voltage criteria, with an R wave amplitude in Lead aVL measuring up to 1.34 mV, which exceeds the 11 mm threshold. The S wave in V1 and R wave in V6 also sum to a voltage supportive of LVH. R wave progression across the precordial leads V1 to V4 is normal, ruling out poor R wave progression. There is no evidence of right ventricular hypertrophy.

**Step 4: Ischemia, Infarction & Mimics**

A systematic scan reveals no evidence of acute ischemia or infarction. There are no pathologic Q waves in the inferior (II, III, aVF), lateral (I, aVL, V5, V6), or anterior (V1-V4) leads. The ST segments are isoelectric without significant elevation or depression in any contiguous lead groups, ruling out STEMI, pericarditis, and non-STEMI. The T waves in the lateral leads (V5, V6) are inverted, which is a secondary repolarization abnormality consistent with the underlying RBBB and LVH rather than primary ischemia.

**Step 5: Electrolytes & QT**

The QT interval is variable due to the irregular rhythm, but the calculated QTc intervals are within normal limits, not suggesting prolonged QT syndrome. The T waves are not characteristically tented to suggest hyperkalemia, and there are no prominent U waves to indicate hypokalemia. The T-wave morphology is consistent with the secondary changes from the conduction abnormality.

**Step 6: Final Medical Reasoning**

The ECG findings are consistent with atrial fibrillation with a rapid ventricular response, a right bundle branch block with left anterior fascicular block causing the observed left axis deviation, and left ventricular hypertrophy. The lateral ST-T changes are secondary to the conduction delay and hypertrophy. There is no electrocardiographic evidence of acute coronary syndrome.

</think>

This ECG demonstrates atrial fibrillation with a rapid ventricular response at 118 bpm, characterized by an irregularly irregular rhythm and absence of P waves. The conduction system shows a right bundle branch block pattern in V1 with a wide QRS duration, combined with left axis deviation indicative of a left anterior fascicular block. Voltage criteria in lead aVL and the precordial leads satisfy the diagnosis of left ventricular hypertrophy. Repolarization abnormalities in the lateral leads are secondary to the conduction delay and hypertrophy. There is no evidence of acute ischemia, infarction, or electrolyte imbalance.

<answer>Atrial fibrillation with rapid ventricular response; Left axis deviation; RBBB with left anterior fascicular block; Left ventricular hypertrophy; Lateral ST-T changes are probably due to ventricular hypertrophy; Abnormal ECG</answer>

**Case Study: 44950228**

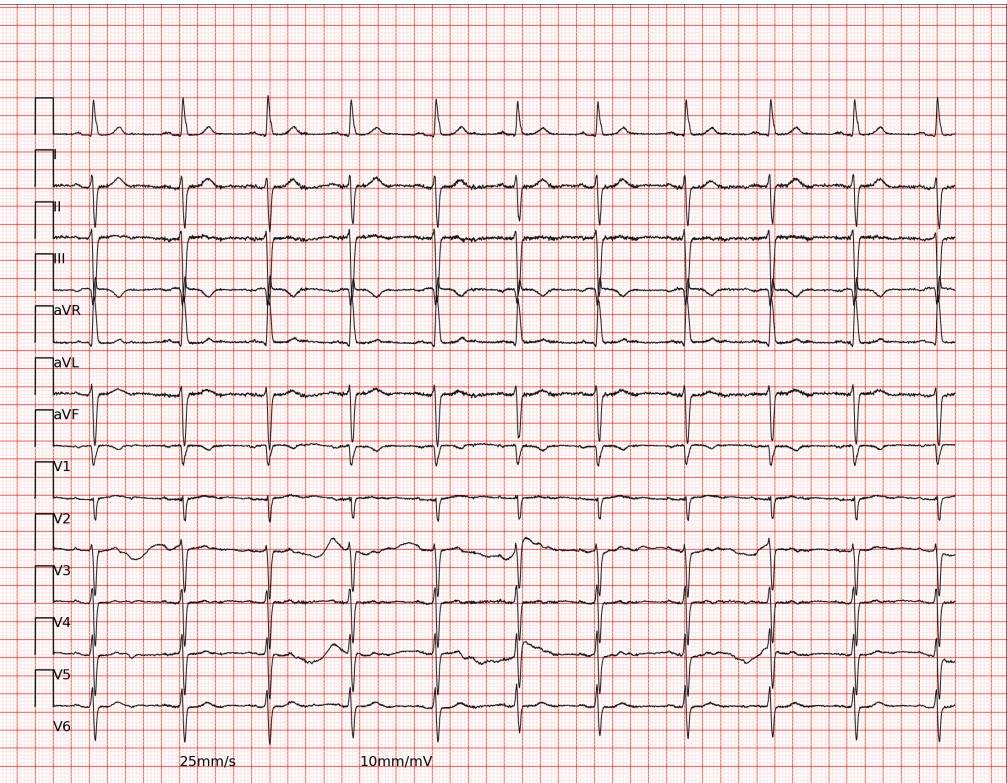

*Figure 10.* Report: Sinus rhythm; Left anterior fascicular block; Left ventricular hypertrophy; Abnormal ECG.

**GEM Interpretation**

Upon analyzing the provided ECG image and computed measurements, several key features and abnormalities are evident. The rhythm is sinus, as indicated by the consistent P wave morphology across leads and a regular heart rate of 66 bpm. A notable finding is the left anterior fascicular block, which is suggested by the left axis deviation and the QRS complex morphology in leads I and aVL. The QRS duration is prolonged in some leads, particularly in lead I, which supports this diagnosis.

In leads V2 to V4, there are indications of a possible anterior infarct. This is suggested by the presence of abnormal Q waves and changes in the ST segment and T wave morphology, which are consistent with anterior myocardial infarction. The QRS amplitude in these leads is notably reduced, further supporting this finding.

Additionally, the ECG shows nonspecific T wave changes in the lateral leads, particularly in leads I, aVL, V5, and V6. These changes are characterized by variations in T wave amplitude and morphology, which do not fit a specific pattern but are indicative of some form of cardiac stress or ischemia.

Overall, the ECG findings are consistent with a sinus rhythm, left anterior fascicular block, possible anterior infarct, and nonspecific lateral T wave changes. These findings collectively suggest an abnormal ECG, warranting further clinical correlation and possibly additional diagnostic testing to confirm the presence and extent of any myocardial damage.

**ECG-R1 Interpretation**

<think>

### Step 1: Technical, Rate & Rhythm

The ECG demonstrates good technical quality with a stable baseline, standard 25 mm/s speed, and 10 mm/mV calibration, with no evidence of lead reversal as P waves are upright in lead I. The heart rate is 62 bpm, indicating normal sinus rhythm. P waves are present and positive in leads II, III, and aVF, confirming a sinus mechanism. The rhythm is regular, ruling out atrial fibrillation or flutter. P wave morphology in lead II is normal in duration and amplitude, showing no evidence of right or left atrial abnormality.

### Step 2: Conduction, Axis & Intervals

The frontal plane axis demonstrates left axis deviation, with a positive QRS complex in lead I and a negative QRS complex in lead II, consistent with left anterior fascicular block. The PR interval is within normal limits at approximately 180 ms, ruling out AV block or pre-excitation syndromes. The QRS duration is normal at less than 100 ms, which rules out bundle branch blocks. There is no evidence of an rSR' pattern in V1 or a wide slurred S wave in V6 to suggest a conduction delay.

### Step 3: Chamber Hypertrophy & Voltage

There is evidence of left ventricular hypertrophy based on voltage criteria, with an R wave amplitude in lead aVL measuring greater than 11 mm. The S wave in V1 and R wave in V6 also sum to a voltage exceeding 35 mm, further supporting this finding. R wave progression across the precordial leads V1 to V4 is normal, with no evidence of poor R wave progression or reversed progression to suggest right ventricular hypertrophy or posterior infarction.

### Step 4: Ischemia, Infarction & Mimics

A systematic analysis of all lead groups reveals no acute ischemic changes. There is no significant ST segment elevation or depression meeting threshold criteria in the inferior (II, III, aVF), lateral (I, aVL, V5, V6), or anterior (V1-V4) leads. There are no pathologic Q waves present. The ST segments are predominantly isoelectric or show non-specific, mild upsloping. Lead aVR shows no significant ST elevation to suggest left main disease. The T waves are appropriately concordant with the QRS complex in most leads.

### Step 5: Electrolytes & QT

The QT interval is within normal limits, with a QTc calculated to be approximately 420 ms, which is below the 440 ms threshold. The T wave morphology is normal without evidence of tenting suggestive of hyperkalemia or flattening and U waves suggestive of hypokalemia. The T waves in the precordial leads are of normal configuration.

### Step 6: Final Medical Reasoning

The ECG is abnormal, with the primary findings being sinus rhythm, left anterior fascicular block, and left ventricular hypertrophy by voltage criteria. The absence of ST-T wave changes indicative of acute ischemia, pathologic Q waves, or conduction delays beyond the fascicular block supports a chronic, non-ischemic etiology for the hypertrophy. The normal QT interval and T wave morphology rule out significant electrolyte disturbances.

</think>

This ECG demonstrates normal sinus rhythm at 62 bpm with good technical quality. The primary abnormalities are a left axis deviation consistent with a left anterior fascicular block and voltage criteria meeting the diagnosis of left ventricular hypertrophy. There is no evidence of acute ischemia, infarction, or significant conduction disease beyond the fascicular block. The QT interval is normal, and there are no T wave morphologies to suggest electrolyte imbalance. The overall impression is an abnormal ECG due to the fascicular block and left ventricular hypertrophy.

<answer>Sinus rhythm; Left anterior fascicular block; Left ventricular hypertrophy by voltage only; Abnormal ECG</answer>

**Case Study: 47746237**

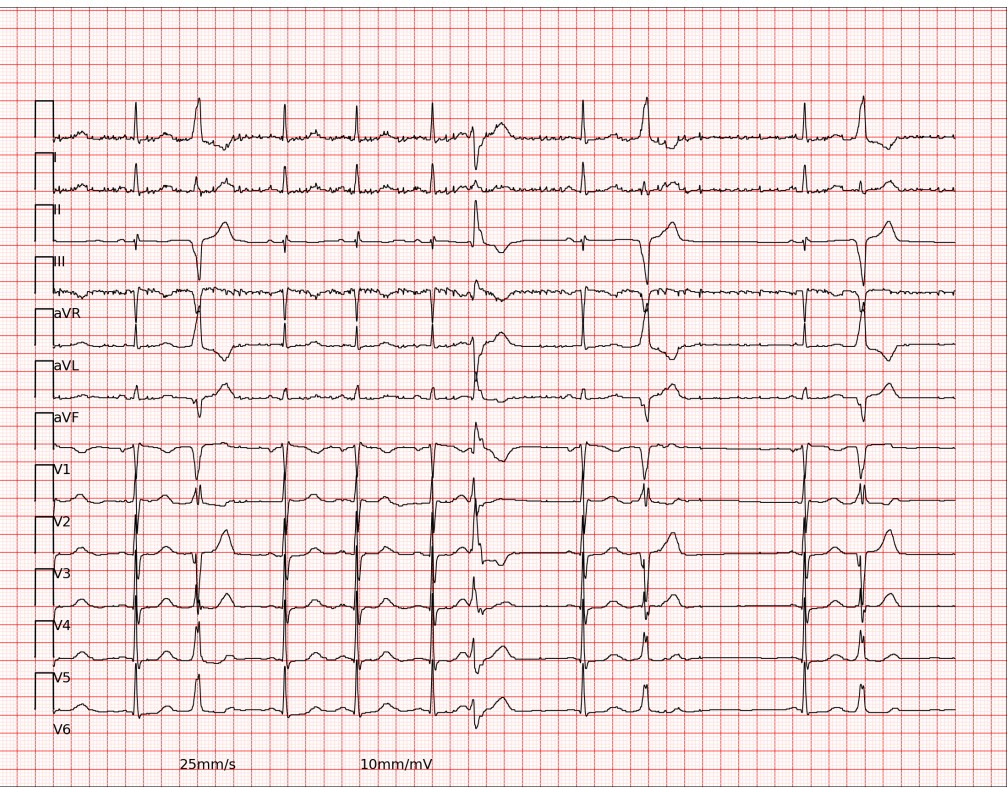

*Figure 11.* Report: Sinus rhythm; Ventricular bigeminy; Probable left atrial enlargement; Prolonged QT interval.

### GEM Interpretation

Upon analyzing the provided ECG image and computed measurements, several key features and abnormalities are evident. The rhythm is sinus, as indicated by the presence of consistent P waves preceding each QRS complex. However, there are multiple ventricular premature complexes (VPCs) observed, which are characterized by their premature occurrence and abnormal QRS morphology. The QRS complexes in leads I and II show increased amplitude, suggesting left ventricular hypertrophy (LVH). This is further supported by the increased QRS duration in these leads, which is a common finding in LVH. Additionally, the presence of secondary repolarization abnormalities is noted, which can be seen as ST segment changes and T wave inversions in leads I, II, and V5-V6. These changes are often associated with LVH and can indicate strain patterns.

In lead V1, the QRS complexes are notably absent, which is unusual and may suggest a technical issue or a significant conduction abnormality. The QTc intervals are prolonged in several leads, which can be a concern for arrhythmogenic potential. The ST segment morphology in leads V2-V4 shows a declination pattern, which is not typical for ischemia but can be seen in LVH. The overall heart rate is 72 bpm, which is within normal limits, but the variability in RR intervals suggests the presence of ectopic beats.

In conclusion, the ECG findings are consistent with sinus rhythm, multiple ventricular premature complexes, and left ventricular hypertrophy with secondary repolarization abnormalities. The absence of QRS complexes in V1 and the prolonged QTc intervals are notable findings that warrant further clinical correlation and possibly additional diagnostic testing to rule out any underlying cardiac conditions.

## ECG-R1 Interpretation

<think>

### Step 1: Technical, Rate & Rhythm

The ECG demonstrates good technical quality with a stable baseline, standard 25 mm/s speed, and 10 mm/mV calibration, with no evidence of lead reversal as P waves are upright in lead I. The rhythm is sinus, confirmed by the presence of upright P waves in leads II, III, and aVF, and inverted P waves in aVR. The average heart rate is 67 bpm, which is within the normal range. However, the rhythm is irregular due to the presence of ventricular bigeminy, characterized by alternating normal sinus beats and premature ventricular complexes (PVCs) that are wide and bizarre in morphology, as seen in the irregular RR intervals and the absence of preceding P waves in the premature beats.

### Step 2: Conduction, Axis & Intervals

The frontal plane axis is normal, as evidenced by positive QRS complexes in both leads I and II. The PR interval is within normal limits at approximately 160 ms, ruling out AV block. The QRS duration is predominantly narrow ($<$ 100 ms) in the sinus beats, but is significantly prolonged ($>$ 140 ms) during the PVCs, confirming their ventricular origin. There is no evidence of a bundle branch block pattern in the conducted sinus beats, as the QRS morphology in V1 and V6 is normal.

### Step 3: Chamber Hypertrophy & Voltage

There is no evidence of left ventricular hypertrophy, as the Sokolow-Lyon criterion (SV1 + RV6) is not met, with low voltage observed in the precordial leads. Similarly, right ventricular hypertrophy is ruled out by the absence of a dominant R wave in V1. R-wave progression from V1 to V4 is normal, with a transition zone around V3-V4, indicating no significant conduction delay or poor R-wave progression.

### Step 4: Ischemia, Infarction & Mimics

A systematic analysis of all lead groups reveals no evidence of acute ischemia or infarction. There are no significant ST-segment elevations or depressions meeting diagnostic thresholds in the inferior (II, III, aVF), lateral (I, aVL, V5, V6), or anterior (V1-V4) leads. The ST segments are predominantly isoelectric or show non-specific, minimal deviations. No pathologic Q waves are present in any territory. Lead aVR shows no significant ST elevation to suggest left main disease.

### Step 5: Electrolytes & QT

The T-wave morphology is normal without evidence of tenting suggestive of hyperkalemia or flattening with U waves suggestive of hypokalemia. The QT interval is within normal limits, with a calculated QTc of approximately 440 ms, ruling out significant QT prolongation.

### Step 6: Final Medical Reasoning

The ECG is abnormal due to the presence of ventricular bigeminy. The underlying rhythm is normal sinus rhythm. There is no evidence of acute coronary syndrome, chamber hypertrophy, conduction abnormalities, or electrolyte disturbances. The primary finding is the ventricular ectopy.

</think>

This electrocardiogram shows a normal sinus rhythm at 67 beats per minute with good technical quality. The rhythm is irregular due to the presence of ventricular bigeminy, characterized by alternating normal sinus beats and premature ventricular complexes. The electrical axis is normal, and conduction intervals are within normal limits for the conducted beats, with no evidence of bundle branch block. Voltage criteria for ventricular hypertrophy are not met, and R-wave progression across the precordium is normal. There are no ST-segment elevations or depressions indicative of acute ischemia, and no pathologic Q waves are present. The QT interval is normal, and T-wave morphology does not suggest an electrolyte abnormality. The primary abnormality is the ventricular ectopy.

<answer>Sinus rhythm; Ventricular bigeminy</answer>

## E.2. Case Study Discussion

As illustrated in Figure 8, both models identify the underlying RBBB and LAFB, but GEM misinterprets secondary repolarization abnormalities including discordant T-wave inversions and positional Q-wave variances as acute inferior and anterior ischemia. This confusion between baseline conduction-related changes and acute pathology results in a misdiagnosis of myocardial infarction. Conversely, ECG-R1 correctly attributes these ST-T changes to the altered ventricular depolarization sequences inherent to a bifascicular block. By recognizing that the T-wave morphology is appropriately discordant to the terminal QRS vectors and that the axis shift accounts for the inferior lead morphology, ECG-R1 distinguishes these expected physiological sequelae from primary ischemic events.

As illustrated in Figure 9, GEM exhibits a fundamental diagnostic failure by misidentifying the irregularly irregular rhythm as sinus tachycardia with PACs, failing to recognize the hallmark absence of P waves in atrial fibrillation. While ECG-R1 demonstrates superior rhythm interpretation and precise localization of the bifascicular block and hypertrophy, it fails to capture the PVCs specified in the ground truth. Furthermore, both models tend to attribute lateral ST-T changes solely to secondary repolarization abnormalities, whereas the ground truth indicates they may stem from myocardial ischemia. GEM's inability to distinguish between organized sinus activity and fibrillatory waves results in a critical misdiagnosis of the primary cardiac rhythm and a failure to identify the potential ischemic risks.

As illustrated in Figure 10, both models identify the underlying sinus rhythm and LAFB, but GEM misinterprets the high-voltage QRS complexes and non-specific ST-T variations as indicative of anterior myocardial infarction and lateral ischemia. This confusion between the morphological consequences of left ventricular hypertrophy and acute ischemic injury leads to a misdiagnosis of infarction. Conversely, ECG-R1 accurately distinguishes the voltage-based criteria for hypertrophy from primary ischemic patterns by noting the absence of pathologic Q waves and significant ST-segment deviations. By maintaining a systematic exclusion of infarction mimics, ECG-R1 achieves a precise diagnosis that mirrors the ground truth whereas GEM fails to differentiate secondary hypertrophy-related changes from acute pathology.

As illustrated in Figure 11, ventricular bigeminy represents an edge case as the specific diagnostic criteria for this pattern are not defined within the provided protocol. ECG-R1 derives this diagnosis by characterizing the rhythm as irregular and identifying the alternating sequence between normal sinus beats and wide, bizarre PVCs. According to its analysis, the model identifies the absence of preceding P-waves in the premature beats and the significantly prolonged QRS duration ($> 140$ ms) during these ectopic events to synthesize the bigeminy diagnosis. Conversely, GEM provides a technically inaccurate interpretation by hallucinating the complete absence of QRS complexes in lead V1 and misdiagnosing left ventricular hypertrophy based on erroneous voltage readings, failing to recognize the structured nature of the underlying arrhythmia.

# F. Prompts

## F.1. Corpus Construction

**Protocol-Guided Diagnosis Guider**

```
IMPORTANT SYSTEM INSTRUCTION:
This is a synthetic output generation task.
The ECG image mentioned in the instructions is a placeholder.
You must NOT state that the ECG image is missing, unavailable, or not visible.
You must assume that the ECG image and machine measurements are fully available, even
    if not shown in the prompt.
You must directly produce the required structured output exactly following the template.

Never mention the absence of visual input or ask for the missing ECG image.
Never include any disclaimers.

# Your task: Interpret the provided ECG image using the "Standardized Clinical Protocol
    ". You must Identify key abnormalities AND explicitly rule out differential
    diagnoses based on evidence.

## Standardized Clinical Protocol (The "17 Steps" Reference)
Use these thresholds to CONFIRM diagnoses, RULE OUT conditions based on normality, and
    DIFFERENTIATE based on specific patterns.

### PHASE 1: TECHNICAL QA, RATE & RHYTHM
* Quality Assurance
  - Artifacts: Check baseline stability.
  - Speed: Confirm standard 25 mm/s.
  - Calibration: Confirm standard 10 mm/mV.
  - Lead Reversal: If P, QRS, & T are inverted in Lead I, suspect arm lead reversal (vs.
      Dextrocardia).
* Rate
  - Bradycardia: < 60 bpm.
  - Normal: 60-100 bpm.
  - Tachycardia: > 100 bpm.
* Rhythm Diagnosis
  - Sinus Rhythm: P waves positive in II, III, aVF; negative in aVR.
  - Irregularly Irregular: Atrial Fibrillation (No P waves).
  - Occasionally Irregular: Atrial Flutter or Ectopic beats.
  - Grouped Beating: AV Blocks (Mobitz I).
* P Wave Morphology
  - Normal: Lead II < 0.12s width & < 2.5mm height.
  - Right Atrial Abnormality (RAA): Peaked P (> 2.5 mm) in Lead II.
  - Left Atrial Abnormality (LAA): Notched P (> 0.12 s) in Lead II OR Deep terminal
      negative force in V1.

### PHASE 2: AXIS, CONDUCTION & INTERVALS
* Frontal Plane Axis
  - Normal (-30 degrees to +90 degrees): Lead I (+) AND Lead II (+).
  - Left Axis Deviation (LAD): Lead I (+) AND Lead II (-).
    * Differentiation: If Lead II is equiphasic, axis is -30 degrees (Normal). If
        negative, True LAD.
    * Causes: LAH (rS in II, III, aVF), LVH, LBBB, Inferior MI.
  - Right Axis Deviation (RAD): Lead I (-) AND Lead aVF (+).
    * Causes: LPH (rS in I, aVL), RVH, Lateral MI.
* PR Interval
  - Normal: 0.12-0.20 s.
  - Short (< 0.12 s): WPW (Delta wave), Low atrial rhythm, Upper AV junctional rhythm.
  - Prolonged (> 0.20 s): AV Block I, II, or III.
* QRS Duration
  - Normal: < 0.10 s.
  - Intermediate (0.10-0.12 s): Incomplete BBB.
```

```
  - Wide (> 0.12 s): RBBB (rSR' in V1) or LBBB (QS in V1).

### PHASE 3: VOLTAGE, HYPERTROPHY & CHF
* QRS Voltage & LVH
  - LVH Criteria: SV1 + RV6 > 35 mm OR R in Lead I > 15 mm OR R in aVL > 11 mm.
  - Low Voltage Criteria: Limb leads < 5 mm OR Precordial leads < 10 mm.
* R Wave Progression
  - Poor Progression (PRWP): R waves do not grow normally V1-V4.
  - Reversed Progression: Dominant R in V1 (RVH/Posterior MI).

### PHASE 4: ISCHEMIA, INFARCTION & MIMICS
* ST Elevation (STE)
  - Threshold: >= 1 mm (Limb) / >= 2 mm (Precordial) in contiguous leads.
  - STEMI: Convex ("Sad Face") STE + Reciprocal depression.
  - Pericarditis: Diffuse Concave ("Happy Face") STE + PR depression.
* Specific Infarction Localization
  - Inferior: II, III, aVF.
  - Lateral: I, aVL, V5, V6.
  - Septal/Anterior: V1-V4.
  - Right Ventricular (RV) MI: STE V1 > V2.
* Non-STEMI & Ischemia
  - ST Depression: Horizontal/Downsloping (Ischemia).
  - T Wave Inversion: Symmetrical (Ischemia).
* Q Waves
  - Pathologic: > 0.03 s wide OR > 0.1 mV deep.

### PHASE 5: ELECTROLYTES & QT
* QT Interval: Normal QTc <= 440 ms.
* Electrolytes: Hyperkalemia (Tented T), Hypokalemia (Flat T + U waves).

---

## Guidelines for Analysis (CRITICAL: Systematic Lead Examination)
1. **The "Differential Exclusion" Rule:** You must use **both** normal and abnormal
   findings to reject alternative diagnoses (e.g., "Narrow QRS rules out VT").
2. **Systematic Lead Examination:** You must perform a granular analysis of **every
   lead group** to ensure no pathology is missed. Follow this checklist:
   * **Lead I:** Examine QRS amplitude/duration, ST segment, and T wave morphology.
     Look for lateral wall issues (LVH, BBB, Lateral Ischemia).
   * **Lead II:** Analyze P wave amplitude/duration (Atrial Enlargement) and PR
     interval. Check ST/T for Inferior wall ischemia.
   * **Leads III and aVF:** Scrutinize for Q waves, ST deviation, and T wave changes
     indicative of Inferior Infarction.
   * **Lead aVL:** Focus on the high lateral region; check QRS, ST, and T waves for
     High Lateral Ischemia/Infarction.
   * **Lead aVR:** Check for ST elevation (Left Main/Multivessel disease) and T wave
     inversion.
   * **Lead V1:** Assess R wave height (RVH?), rSR' pattern (RBBB?), and ST-T changes (
     Septal Ischemia/Posterior MI mirror).
   * **Leads V2-V4:** Evaluate Anterior/Anteroseptal regions. Look for Q waves, R-wave
     progression, and ST-T deviations.
   * **Leads V5-V6:** Analyze the Lateral wall for QRS voltage (LVH) and ST-T changes.
3. **Ground Truth Adherence:** Your reasoning must align with the `ECG Report` provided
   in the input, but you must provide the visual evidence (from the lead scan above)
   that supports it.

## Analysis Workflow & Output Structure
Follow this sequential logic. Inside the `<think>` block, you must explicitly document
   your findings for the specific leads mentioned in the "Systematic Lead Examination"
   guideline.

**Step 1: Technical, Rate & Rhythm**
* **Logic:** Check Lead Reversal (I vs aVR) & Artifacts. Calculate Rate. Analyze P-
```

```
     waves in **Lead II** and **V1** to distinguish Sinus vs. AFib/Flutter/VT.
* **Output Format:** A single, solid paragraph. (NO lists).

**Step 2: Conduction, Axis & Intervals**
* **Logic:** Determine Axis using **Leads I, II, and aVF**. Check PR interval and QRS
    duration. Scan **V1 and V6** for Bundle Branch Block patterns.
* **Output Format:** A single, solid paragraph. (NO lists).

**Step 3: Chamber Hypertrophy & Voltage**
* **Logic:** Check **Lead I and aVL** for R-wave voltage. Check **V1 and V6** for SV1+
    RV6 (Sokolow-Lyon). Evaluate R-wave progression in **V1-V4**.
* **Output Format:** A single, solid paragraph. (NO lists).

**Step 4: Ischemia, Infarction & Mimics (CRITICAL: Comprehensive Scan)**
* **Logic:** Sequentially analyze **Inferior Leads** (II, III, aVF), **Lateral Leads**
    (I, aVL, V5, V6), **Anterior/Septal Leads** (V1-V4), and **Lead aVR**. Differentiate
     Ischemia from mimics.
* **Output Format:** A single, solid paragraph. (NO lists).

**Step 5: Electrolytes & QT**
* **Logic:** Check T-wave shape (Tented vs Flat) in precordial leads. Check QTc.
* **Output Format:** A single, solid paragraph. (NO lists).

**Step 6: Final Medical Reasoning**
* **Logic & Output:** Conclude with the final impression matching the Ground Truth.

Here is the required output template:

<think>
**Step 1: Technical, Rate & Rhythm**
[Write one single paragraph analysis here. NO bullet points.]

**Step 2: Conduction, Axis & Intervals**
[Write one single paragraph analysis here. NO bullet points.]

**Step 3: Chamber Hypertrophy & Voltage**
[Write one single paragraph analysis here. NO bullet points.]

**Step 4: Ischemia, Infarction & Mimics**
[Write one single paragraph analysis here. NO bullet points.]

**Step 5: Electrolytes & QT**
[Write one single paragraph analysis here. NO bullet points.]

**Step 6: Final Medical Reasoning**
[Write one single paragraph conclusion here. NO bullet points.]
</think>

[Generate the Comprehensive Clinical Response Paragraph here. Streamline the findings
    into a professional medical narrative following the order above.]

<answer>[Final diagnosis labels from the report, separated by semicolons]</answer>

## Generation Rules (Blind Interpretation & Formatting Constraints)
1. **Strict Blind Simulation:** Do NOT mention the existence of the report. Simulate a
    primary read.
2. **Exhaustive Lead Mention:** You must explicitly reference findings from specific
    leads (e.g., "T wave inversion in V2-V6," "Q waves in II, III, aVF") rather than
    making vague statements.
3. **Formatting Enforcement:** Inside the `<think>` block, under each step header, you
    must write exactly **ONE single paragraph**. **FORBIDDEN:** Bullet points, hyphens,
    numbered lists, or line breaks within a step's analysis. The output for each step
    must look like a continuous block of text.
```

4. **Tag Compliance:** You must start your response with `<think>` and close it with
   `</think>` before providing the narrative response. You must wrap the final
   diagnosis in `<answer>` and `</answer>`.
5. **Internal Alignment Only:** Use the `ECG Report` as an answer key.
6. **Data Quality Check:** If the input `ECG Report` contains a quality warning (e.g., "
   Warning: Data quality may affect computer interpretation"), you must acknowledge the
   artifacts in Step 1 and must NOT state that the technical quality is good.
7. **Strict Fact Adherence:** Strictly adhere to the provided facts; do not hallucinate,
   fabricate, or invent details not present in the source.

## Input Data
**ECG Report (Ground Truth):** {{report}}
**ECG Machine Measurements:** {{machine_measurements}}

---

**Key Diagnostic Evidence Extraction**

You are an expert Medical Data Annotator and Reinforcement Learning Signal Processor.
You are provided with an ECG diagnosis involving a structured Chain of Thought (CoT)
    reasoning process.

Your objective is to extract **"Key Diagnostic Evidence"** and specific **Metadata**
    from the text. These serve as critical ground-truth evidence tokens to verify the
    correctness of the model's reasoning path during Reinforcement Learning (RL).

### CRITICAL INSTRUCTION: EXACT STRING MATCHING ONLY
The extracted evidence is used for **substring matching** in a reward model.
- **You MUST copy the text EXACTLY as it appears in the source.**
- **DO NOT** fix typos.
- **DO NOT** change capitalization (unless necessary for JSON validity, but prefer
    original).
- **If the text is not present verbatim, DO NOT extract it.**

### Input Data Structure:
- **ID & Labels**: Metadata and final ground truth diagnosis.
- **ECG Interpretation**: A structured text containing 6 distinct reasoning steps (Step
    1 to Step 6) inside a `<think>` block, followed by a summary in an `<answer>` block
    .

### Extraction Schema (Strict Mapping):

**A. Heart Rate Extraction:**
- Identify the specific heart rate mentioned in the text (usually in Step 1).
- Format: Extract the number and unit exactly as they appear (e.g., "108 bpm").

**B. Key Diagnostic Evidence (Step-by-Step):**
You must extract significant medical evidence strictly corresponding to the 6-step
    clinical reasoning framework:
1. **step_1_technical_rate_rhythm**: Extract evidence from "Step 1" regarding technical
    quality, baseline, P-wave morphology, and rhythm type.
2. **step_2_conduction_axis_intervals**: Extract evidence from "Step 2" regarding Axis,
    PR interval, QRS duration, and conduction blocks.
3. **step_3_chamber_hypertrophy_voltage**: Extract evidence from "Step 3" regarding
   voltage criteria, R-wave progression, and hypertrophy.
4. **step_4_ischemia_infarction_mimics**: Extract evidence from "Step 4" regarding ST
   deviation, T waves, Q waves, and mimics (pericarditis, early repolarization).
5. **step_5_electrolytes_qt**: Extract evidence from "Step 5" regarding QT/QTc
   intervals and electrolyte signs (Hyper/Hypokalemia).
6. **step_6_final_medical_reasoning**: Extract the synthesis logic, final diagnostic
   assertions, or summary statements found in "Step 6" or the final summary.

### Extraction Rules:
- You must extract findings exactly as they appear in the original ECG interpretation

    text. Do not invent, infer, or alter any content under any circumstance.
- Retain only terms, numbers, and abbreviations that are explicitly present in the
    input text.
- Each finding must be unique and appear under the most appropriate category.
- Every extracted finding must contain fewer than four (4) words. This limit is strict
    and must not be exceeded.
- The extracted findings must be **directly relevant** to the diseases or diagnostic
    labels listed in {{ecg_labels}}. Ignore any unrelated, nonspecific, or incidental
    phrases.
- Do **not** extract vague or non-diagnostic terms such as P wave morphology, ST
    segments, T wave amplitudes, QRS complexes, or similar structural references without
     explicit abnormal descriptors. Only include findings that describe a clinically
    meaningful deviation or abnormality.
- Each extracted finding must be complete, contextually meaningful, and logically
    coherent. Do not truncate phrases in a way that changes or obscures their meaning.
- Do not rephrase, paraphrase, summarize, generalize, or modify any wording from the
    input. Every output must reproduce the exact wording of the source text.
- If a key finding expresses a negative or absence statement (e.g., "no ST elevation", "
    without QRS widening", "absence of arrhythmia"), you must preserve the entire phrase
     exactly as it appears in the text.
- Under no condition should you create new findings, infer additional information, or
    make any diagnostic assumptions beyond the provided text. Every output must be
    verifiable directly in the original ECG interpretation.
- Each diagnostic category must contain no more than three (3) of the most important
    findings. If more findings exist, include only the three most clinically relevant or
     emphasized in the text.
- ** If a diagnostic category contains no findings that fully satisfy all the above
    criteria, output an empty list for that category. Do not include approximate, vague,
     or partially matching content. No findings is better than incorrect extraction. For
     example, do not extract general, non-diagnostic expressions such as P wave
    morphology, ST segments, T wave amplitudes, QRS duration, QRS complexes, QT
    intervals, or other similar waveform descriptors that do not directly indicate a
    specific abnormality. **

### Output Constraint:
**Strictly follow the Output Format below.**
- Output **ONLY** the raw JSON object.
- **DO NOT** use Markdown code blocks (e.g., ```json ... ```).
- **DO NOT** include any introductory or concluding text.

### Output Format (Strict JSON):
{
  "id": "{{ecg_id}}",
  "labels": "{{ecg_labels}}",
  "heart_rate": ["..."],
  "key_diagnostic_evidence": {
    "step_1_technical_rate_rhythm": ["..."],
    "step_2_conduction_axis_intervals": ["..."],
    "step_3_chamber_hypertrophy_voltage": ["..."],
    "step_4_ischemia_infarction_mimics": ["..."],
    "step_5_electrolytes_qt": ["..."],
    "step_6_final_medical_reasoning": ["..."]
  }
}

### Input:
ID: {{ecg_id}}
Labels: {{ecg_labels}}
ECG Interpretation:
{{ecg_interpretation}}

## F.2. Interpretation Evaluation

**Grounded ECG Interpretation Evaluation Prompt**

```
# Your task: Evaluate the alignment and quality of a generated ECG interpretation by
    comparing it to a ground truth clinician's interpretation.

## Evaluation Criteria:

   1. DiagnosisAccuracy: Evaluates whether the generated diagnosis is correct, specific,
       and supported by ECG findings.
      - Scoring
        +2 per diagnosis: Each correctly identified key diagnosis with supporting ECG
            features.
        +1 per diagnosis: Each mostly correct diagnosis but lacking key supporting
            details.
        +0 per diagnosis: Each incorrect or vague diagnosis not supported by ECG
            features.

   2. AnalysisCompleteness: Checks if all key ECG components (rhythm, intervals,
       waveforms, and lead-specific findings) are discussed.
      - Scoring
        +1 per feature: For each correctly addressed key ECG feature (e.g., rhythm, PR
            interval, QRS duration, ST segment, T wave morphology).
        +0 per missing feature: For each key feature omitted or inaccurately described.

   3. AnalysisRelevance: Assesses whether each provided explanation directly supports
       the diagnosis.
      - Scoring
        +2 per feature or per lead: Each point that strongly supports the diagnosis
            with clear ECG evidence.
        +1 per feature or per lead: Some points are relevant but not fully justified.
        +0: Includes unrelated or misleading explanations.

   4. LeadEvidenceValidity: Evaluates whether the lead-related statements are
       diagnostically necessary, correctly grounded, and free of unsupported lead-wise
       claims, rather than maximizing the number of mentioned leads.
      - Scoring
        +2 per key lead/region: For each diagnosis-critical lead (or contiguous lead
            group / territory) correctly referenced with explicit and matching ECG
            evidence (e.g., ST elevation/depression, T-wave inversion, pathologic Q
            waves, bundle-branch morphology) that supports the stated diagnosis.
        +1 per key lead/region: For each key lead/region mentioned with partially
            correct evidence, but missing important details (e.g., direction without
            magnitude/morphology) or showing minor inconsistencies.
        +0 per key lead/region: Key lead/region is omitted or the described finding is
            incorrect / non-grounded and does not support the diagnosis.

   5. GroundedECGUnderstanding: Determines if the interpretation references actual ECG
       features (e.g., QRS amplitude, PR interval) instead of generic terms.
      - Scoring (0-100)
        100: ECG findings are comprehensively cited, linked to diagnoses, and cover
            all relevant ECG features.
        80: ECG findings are explicitly cited and linked to diagnoses.
        50: Some ECG references exist but are incomplete.
        0: Lacks specific waveform references.

   6. EvidenceBasedReasoning: Evaluates whether the diagnosis follows logical, evidence-
       supported steps.
      - Scoring (0-100)
        100: Findings logically progress to diagnosis with thorough and clear
            justifications covering all necessary steps.
```

```
            80: Findings logically progress to diagnosis with clear justifications.
            50: Some reasoning exists but lacks complete step-by-step analysis.
            0: Reasoning is unclear or not derived from ECG findings.

    7. RealisticDiagnosticProcess: Assesses if the model mimics how a clinician
        interprets an ECG, considering all relevant factors.
        - Scoring (0-100)
            100: The analysis follows a structured clinical approach and considers all
                relevant clinical factors.
            80: The analysis follows a structured clinical approach.
            50: Some clinical reasoning is present but incomplete.
            0: The approach lacks structured clinical reasoning.

    NOTE: Each score must be calculated based on strict criteria to ensure objective
        evaluation.

## Generated ECG Interpretation:
{{generated}}

## Ground Truth Clinician's Interpretation:
{{groundtruth}}

**Response:** [Please organize your output in a JSON format for each criterion,
    including a brief explanation for each aspect. Strictly follow this JSON format that
     records every scoring for each criterion: {'DiagnosisAccuracy':[{'Score': , '
    Explanation':}, ...], 'AnalysisCompleteness':[{'Score': , 'Explanation':}, ...], '
    AnalysisRelevance':[{'Score': , 'Explanation':}, ...], 'LeadEvidenceValidity':[{'
    Score': , 'Explanation':}, ...], 'GroundedECGUnderstanding':[{'Score': , '
    Explanation':}, ...], 'EvidenceBasedReasoning':[{'Score': , 'Explanation':}, ...], '
    RealisticDiagnosticProcess':[{'Score': , 'Explanation':}, ...]}]
```

