# OpenReview forum: "ECG-R1: Protocol-Guided and Modality-Agnostic MLLM for Reliable ECG Interpretation"
_ICML.cc/2026/Conference — ICML 2026 regular_

### Official Review · Reviewer_5z62 · 2026-03-09

**Soundness:** 3
**Presentation:** 3
**Significance:** 3
**Originality:** 3
**Overall Recommendation:** 4
**Confidence:** 3

**Summary:**

This paper proposes ECG-R1, a multimodal large language model for reliable ECG interpretation. The method combines three main components: (1) Protocol-Guided Instruction Data Generation, which uses extracted ECG features and monograph-derived diagnostic logic to generate structured interpretation data; (2) a decoupled multimodal architecture with Interleaved Modality Dropout (IMD) to improve robustness and cross-modal consistency when either ECG signals or ECG images are missing; and (3) reinforcement learning with ECG Diagnostic Evidence Rewards (EDER) to encourage evidence-grounded intermediate reasoning in addition to final diagnostic correctness.

**Compliance With Llm Reviewing Policy:**

Affirmed.

**Key Questions For Authors:**

1.How robust are the main conclusions to evaluator choice?

Since the paper relies on automatic model-based evaluation for several metrics, it would be helpful to clarify whether the ranking of ECG-R1 versus GEM and other strong baselines remains stable across different evaluators.

2.Can the authors provide a clearer attribution of gains across the three main components?

The paper presents multiple contributions—protocol-guided data generation, decoupled encoding with IMD, and EDER. A concise summary of which components drive which gains would strengthen the paper.

3.To what extent are the improvements tied to the structured output schema?

Because the model is trained with a protocolized reasoning format, it would be useful to better understand how much of the gain comes from improved ECG understanding versus improved adherence to a structured reporting style.

4.Can the authors more precisely scope the novelty claim around being the “first reasoning MLLM” for ECG interpretation?

A narrower and more explicit statement of novelty would make the positioning relative to prior ECG-focused and medical reasoning models clearer.

**Limitations:**

yes

**Strengths And Weaknesses:**

This is a strong and timely paper on reliable ECG interpretation with multimodal LLMs. The problem is important, and the paper is well motivated throughout. The proposed framework is coherent: protocol-guided data generation, decoupled multimodal encoding with interleaved modality dropout, and evidence-rewarded RL all target clear failure modes in prior work, especially hallucination and brittleness under missing modalities. The empirical results are convincing overall, with substantial gains over prior ECG-specialized baselines and a useful large-scale comparison across proprietary, open-source, medical, and ECG-specific MLLMs. The paper is also clearly written and generally well structured, and I found the method and experimental setup easy to follow.

My main reservations are about claim calibration rather than core technical quality. Although the protocol-guided corpus is more grounded than prior prompting-based approaches, the supervision still depends on LLM-generated reasoning traces, so it is not fully clear how much the model learns faithful ECG reasoning versus a structured reporting style. Relatedly, some evaluation metrics appear naturally aligned with the target output format, which may somewhat favor the proposed method. I also think some novelty claims, especially around being the “first reasoning MLLM,” should be scoped more carefully. Overall, however, these concerns do not outweigh the paper’s strengths: the work is technically solid, practically relevant, and likely to be useful to researchers working on reliable multimodal reasoning in medical settings.

---

> ### Author Rebuttal · Authors · 2026-03-29
>
> We sincerely thank the reviewer for the constructive comments and recognition. We will scope the novelty claims more carefully as suggested. Below, we respond to each concern point by point.
>
>
>
> > [W1 & Q3] The supervision still depends on LLM-generated reasoning traces, and it is not fully clear how much the model learns faithful ECG reasoning versus a structured reporting style.
>
> We thank the reviewer for raising this important point. We acknowledge that fully disentangling faithful ECG reasoning from structured reporting style remains an open challenge with LLM-generated supervision. That said, we would like to draw attention to the Evidence-Based Reasoning metric in Table 1, which evaluates whether the diagnosis follows logical, evidence-supported reasoning steps rather than assessing output format. ECG-R1 achieves 79.08 on this metric compared to GEM's 63.15, which may suggest that the model acquires reasoning capabilities beyond structured reporting. Additionally, our structured protocol is designed to serve clinical needs: it ensures critical checks are not omitted (e.g., QT interval analysis is frequently missed in free-text generation) and enables clinicians to quickly locate specific interpretation sections. We will discuss this further in the revised paper.
>
>
>
> > [W2 & Q1] Some evaluation metrics appear naturally aligned with the target output format, and it would be helpful to clarify whether the ranking remains stable across different evaluators.
>
> We acknowledge that some evaluation metrics (e.g., Analysis Completeness, Clinical Diagnostic Fidelity) may have a natural advantage for structured outputs. However, the core metric Diagnosis Accuracy purely measures diagnostic correctness regardless of output format, and ECG-R1 also leads significantly on this metric (80.29 vs. GEM 74.70 in Table 1).
>
> Regarding evaluator robustness, **we conducted additional evaluations using GLM-5, a latest open-source LLM, as an independent evaluator under the same rubric**, selecting representative models from each category:
>
> | Model               | Diagnosis Accuracy | Analysis Completeness | Analysis Relevance | Lead Evidence Validity | ECG Feature Grounding | Evidence Based Reasoning | Clinical Diagnostic Fidelity |
> | ------------------- | ------------------ | --------------------- | ------------------ | ---------------------- | --------------------- | ------------------------ | ---------------------------- |
> | GPT-5.1-Instant     | 32.24              | 1.86                  | 1.66               | 1.57                   | 56.72                 | 39.65                    | 57.12                        |
> | MiniCPM-V-4.5       | 29.72              | 1.60                  | 1.24               | 0.68                   | 44.36                 | 28.70                    | 46.33                        |
> | HuatuoGPT-Vision-7B | 34.93              | 2.21                  | 1.64               | 0.31                   | 45.85                 | 31.62                    | 40.26                        |
> | PULSE               | 70.68              | 1.87                  | 2.81               | 0.69                   | 50.44                 | 47.32                    | 48.21                        |
> | GEM                 | 79.14              | 3.79                  | 4.45               | **4.61**               | 75.43                 | 69.84                    | 72.05                        |
> | ECG-R1 (Ours)       | **83.86**          | **3.99**              | **4.79**           | 4.58                   | **87.29**             | **81.40**                | **89.03**                    |
>
> While absolute scores differ from DeepSeek due to inherent biases across LLMs, the ranking remains largely consistent — ECG-R1 achieves the best performance across the vast majority of metrics. **Additionally, the expanded Cardiologist Evaluation presented in our response to Reviewer ZHTT [W2], which is entirely independent of any LLM evaluator, consistently rated ECG-R1 as the top-performing model.** These results together confirm that our conclusions are robust to evaluator choice.
>
>
>
> > [W3 & Q4] The novelty claim around being the "first reasoning MLLM" should be scoped more precisely.
>
> We thank the reviewer for this suggestion. We agree that the claim should be scoped more precisely and will revise it to "the first reasoning ECG MLLM" in the updated paper, to more clearly distinguish our positioning from prior general-purpose reasoning models.
>
>
>
> > [Q2] A clearer attribution of gains across the three main components would strengthen the paper.
>
> We thank the reviewer for this helpful suggestion. The ablation results for each component are currently presented separately across Figure 6 (Protocol-Guided CoT data), Tables 5–6 (IMD), and Table 7 (EDER). We will add a dedicated summary section in the revised paper to consolidate the gain attribution of all three components in one place for easier reference.

---

> > ### Author Rebuttal · Reviewer_5z62 · 2026-04-04
> >
> > Thank you to the authors for their detailed rebuttal and clarifications. I have read the response and appreciate the additional explanations provided. After considering the rebuttal, I believe my main concerns have been adequately addressed to the extent possible, but my overall assessment of the paper remains unchanged. Therefore, I decide to maintain my original score.

---

> > > ### Author Response · Authors · 2026-04-05
> > >
> > > We sincerely thank the reviewers for their careful reading of our work and for providing constructive feedback. We have carefully considered all comments and made corresponding improvements in the revised manuscript.

---

### Official Review · Reviewer_QGCx · 2026-03-09

**Soundness:** 3
**Presentation:** 2
**Significance:** 2
**Originality:** 3
**Overall Recommendation:** 4
**Confidence:** 4

**Summary:**

This paper proposed ECG-R1, which is the first reasoning MLLM for ECG interpretation. The frameworks contains three components, including protocol-guided data generation to obtain more reliable reasoning data, interleaved modality dropout to adapt to missing modality scenario, and ECG diagnostic reward for clinical evidence matching. Comparison with baselines demonstrates improved performance.

**Compliance With Llm Reviewing Policy:**

Affirmed.

**Final Justification:**

The authors' reply clarifies the multi-modal fusion mechanism. Furthermore, the analysis provided in the rebuttal resolves my previous questions regarding the measurement scores within the reports. I will adjust my score to 4.

**Key Questions For Authors:**

1. According to Appendix B.2, only LLM is learnable in both SFT and RL stages, while image encoder, ECG encoder, and the projectors are frozen. Can the features from both modalities be aligned with LLM space in this training strategy, instead of just pushing LLM to blindly output language with similar structures to the training data?
2. Figure 5 shows that Grounding CoT data contains exact digits representing the specific measurement in ECG, such as ventricular rate. How accurately can ECG-R1 measure these indicators?

**Limitations:**

Suggest to include a limitation section.

**Strengths And Weaknesses:**

## Strengths

1. Protocol-guided data generation grounds reasoning in clinical guidelines, injecting professional workflows into CoT data.
2. Interleaved modality dropout tackles a significant problem, missing modality, in ECG interpretation of MLLM.
3. Experiments show consistently superior performance of ECG-R1 across LLM-as-a-judge metrics from different aspects.

## Weaknesses

1. Failing to discuss and model the relation and interaction between modalities. In Theorem 2.2, the authors assume that $\Delta_{view}$ and $\Delta_{swap}$ is negligible. However, it is intuitive that image and signal transmit information in different ways, so the feature and noise in both modalities are different. As shown in Appendix C.3, the model input with omni modalities performs better that inputting with missing modality, which also demonstrate that different modalities provide complementary information.
2. Limited multimodal reasoning capacity of reward design. Though the proposed ECG Diagnostic Evidence Reward encourages the model to align with diagnostic evidence, it is mostly limited to text-level matching, instead of text-image or text-signal alignment, which appears more promising to ground observations in reasoning.
3. Fairness of benchmark comparison.  DeepSeek-V3.1-Terminus is applied to both data generation and LLM-as-a-Judge metric evaluation for ECG-R1. The improved results might be attributed to the fact that LLM favors data that align with its generation distribution.

---

> ### Author Rebuttal · Authors · 2026-03-29
>
> We sincerely thank the reviewer for the detailed feedback. Below, we respond to each concern point by point.
>
>
>
> > [W1] Failing to discuss and model the relation and interaction between modalities. In Theorem 2.2, the authors assume that Δview and Δswap is negligible. However, it is intuitive that image and signal transmit information in different ways, so the feature and noise in both modalities are different. As shown in Appendix C.3, the model input with omni modalities performs better that inputting with missing modality, which also demonstrate that different modalities provide complementary information.
>
> We sincerely thank the reviewer for this thoughtful observation, and we apologize for the unclear presentation. **We would like to note that Δview does not measure whether the two modalities have different features or noise — instead, it measures the total variation distance between the Bayes-optimal predictions given each single modality alone.** Intuitively, since both modalities encode the same underlying cardiac electrical activity, examining either in isolation should lead to the same diagnosis — this is precisely what a small Δview means. The same applies to Δswap: reordering modality token blocks should not change the correct diagnosis. We will revise the presentation to make this distinction clearer in the updated paper.
>
>
>
> > [W2] Limited multimodal reasoning capacity of reward design. Though the proposed ECG Diagnostic Evidence Reward encourages the model to align with diagnostic evidence, it is mostly limited to text-level matching, instead of text-image or text-signal alignment, which appears more promising to ground observations in reasoning.
>
> We sincerely thank the reviewer for this insightful suggestion. In our current design, evidence phrases are derived from FeatureDB which extracts measurements directly from ECG signals, so text matching indirectly verifies signal-level consistency. Introducing cross-modal rewards would require an additional differentiable verifier, increasing cost and potential noise. Even with text-level matching alone, EDER improves ECG Feature Grounding from 78.53 to 80.57 (Table 7), demonstrating consistent gains. Inspired by the reviewer's valuable suggestion, we will discuss multimodal reward design as an important future direction in the revised paper.
>
>
>
> > [W3] Fairness of benchmark comparison. DeepSeek-V3.1-Terminus is applied to both data generation and LLM-as-a-Judge metric evaluation for ECG-R1. The improved results might be attributed to the fact that LLM favors data that align with its generation distribution.
>
> We sincerely appreciate the reviewer for raising this important concern. We conducted two additional experiments to address it: **(1) an independent evaluation using GLM-5 as an alternative LLM grader under the same rubric**, and **(2) an expanded cardiologist evaluation. Both experiments yield consistent conclusions** — ECG-R1 remains the top-performing model regardless of the evaluator, confirming alignment between LLM-based and human expert assessments. **Due to character limits, full results are presented in our responses to Reviewer 5z62 [W2 & Q1] and Reviewer ZHTT [W2].** We are grateful that this concern prompted these additional experiments, which further strengthen the paper.
>
>
>
> > [Q1] According to Appendix B.2, only LLM is learnable in both SFT and RL stages, while image encoder, ECG encoder, and the projectors are frozen. Can the features from both modalities be aligned with LLM space in this training strategy, instead of just pushing LLM to blindly output language with similar structures to the training data?
>
> We thank the reviewer for identifying this ambiguity, and we apologize for the unclear description. We would like to note that in the SFT stage, only encoders are frozen while projectors and LLM are trained, completing modality alignment, following the practice of LLaVA. In the RL stage, projectors are also frozen and only the LLM is trained to refine reasoning strategy, consistent with approaches such as VLM-R1. We will revise Appendix B.2 in the updated paper.
>
>
>
> > [Q2] Figure 5 shows that Grounding CoT data contains exact digits representing the specific measurement in ECG, such as ventricular rate. How accurately can ECG-R1 measure these indicators?
>
> We thank the reviewer for raising this important question. We conducted an additional evaluation by extracting measurement values from ECG-R1's generated CoT outputs and comparing them against the interpretation text in the test set.
>
> | Indicator            | MAPE   |
> | -------------------- | ------ |
> | Heart Rate (bpm)     | 4.46%  |
> | PR Interval (ms)     | 16.42% |
> | QT/QTc Interval (ms) | 15.62% |
>
> The results show that ECG-R1 maintains reasonable measurement accuracy across these indicators. We will include this analysis and discuss potential improvements in the limitation and future work section of the revised paper, as suggested.

---

> > ### Author Rebuttal · Reviewer_QGCx · 2026-04-03
> >
> > Thanks the authors for the response. I have following questions regarding the relation between both modalities and the performance of measurement.
> > 1. Regarding W1, the authors state that both modalities encode the same activity, and examining either should lead to the same diagnosis. In this circumstance, what is the advantage of the fusion of both modalities? Whether the better performance of model input with omni modalities derives from complementary information of different modalities or not? If so, how does the model facilitate this interaction between different modalities during training?
> > 2. Regarding Q2, the MAPE for PR Interval and QT/QTc Interval seems large. Could this be attributed to that there lacks supervision of measurement during training?

---

> > > ### Author Response · Authors · 2026-04-05
> > >
> > > We sincerely thank the reviewer for the thoughtful follow-up questions. Below, we address each sub-question (SQ) point by point.
> > >
> > > > [SQ1] What is the advantage of the fusion of both modalities?
> > >
> > > **The advantage of fusion lies in reducing the overall diagnostic error by combining two partially independent feature representations of the same diagnostic truth.**
> > >
> > > Theorem 2.3 provides a direct theoretical explanation: F(θ) ≤ Δview + √(ετI(θ)/2) + √(ετT(θ)/2). Since ECG images and signals are two renderings of the same cardiac waveform, the correct diagnosis does not change with modality, hence Δview is small (defined in Eq. 9). However, the actual model is not Bayes-optimal, and each modality environment carries its own excess risk ετ(θ) (Eq. 10). In practice, the image encoder and signal encoder have different architectural inductive biases and are subject to different noise characteristics, causing ετI(θ) and ετT(θ) to be both non-zero with different error profiles.
> > >
> > > **Therefore, Δview being small provides the prerequisite for meaningful fusion, as both modalities target the same diagnostic truth. At the same time, the different error profiles of ετI(θ) and ετT(θ) indicate that the two encoders extract complementary features from each modality, allowing the model to compensate for each modality's weaknesses using the other.**
> > >
> > >
> > >
> > > > [SQ2] Whether the better performance derives from complementary information of different modalities?
> > >
> > > **Yes. The two encoders extract complementary features due to their different architectures and noise characteristics.** For example, the image encoder better captures morphological patterns (e.g., ST-segment shape), while the signal encoder provides more precise temporal measurements (e.g., interval durations). The omni-modality gain of ~2.4 points in Table 5 is consistent with this complementarity, as fusion allows the model to compensate for each modality's weaknesses.
> > >
> > >
> > >
> > > > [SQ3] How does the model facilitate this interaction between different modalities during training?
> > >
> > > The cross-modal interaction is facilitated through the LLM's self-attention mechanism: signal tokens and image tokens are concatenated into a single sequence, where each token attends to all others regardless of modality origin, enabling cross-modal correlations and a unified diagnostic representation. **IMD (Interleaved Modality Dropout) is critical for shaping genuine interaction: without IMD, the model overfits to the co-presence of both modalities (omni 81.99 but collapses to 36.77 under modality missing, Table 5), rather than truly integrating each modality's information. With IMD, Diagnosis Accuracy remains 77.91 even when one modality is entirely absent, and further improves to 80.29 under omni-modality, confirming robust representations and additional complementary benefit through cross-modal interaction.**
> > >
> > >
> > >
> > > > [SQ4] Regarding Q2, the MAPE for PR Interval and QT/QTc Interval seems large. Could this be attributed to that there lacks supervision of measurement during training?
> > >
> > > We agree with this assessment. **We conducted a statistical analysis of the training corpus and found that heart rate appears with explicit numerical values in 99.8% of samples, whereas PR interval and QT/QTc interval only contain numerical values in 66.6% and 84.1% respectively, with the remaining samples providing only qualitative descriptions or omitting measurements entirely.** This disparity in quantitative supervision directly explains the MAPE gap.
> > >
> > > Furthermore, we conducted a paired comparison to examine whether RL with EDER can partially mitigate this issue. We extracted measurement values from both SFT and RL model outputs and retained only the samples where both models produced explicit numerical values, ensuring the comparison is on exactly the same set of samples (the absolute values differ slightly from our previous response due to this stricter filtering criterion).
> > >
> > > | Indicator       | SFT    | RL         |
> > > | --------------- | ------ | ---------- |
> > > | PR Interval     | 17.56% | **16.57%** |
> > > | QT/QTc Interval | 15.34% | **15.16%** |
> > >
> > > EDER partially compensates by rewarding diagnostic evidence phrases, incentivizing more accurate measurements. We will discuss potential improvements such as numerical regression auxiliary objectives in future work.
> > >
> > >
> > >
> > > We sincerely thank the reviewer for the in-depth discussion. Across two rounds, we hope all concerns have been substantially addressed: the theoretical role of Δview and complementary error profiles (W1, SQ1-SQ2), cross-modal interaction via self-attention and IMD (SQ3), evaluation fairness through independent LLM grader and expanded cardiologist evaluation (W3), frozen vs. trainable components (Q1), multimodal reward as future work (W2), and the MAPE gap explanation with RL improvement evidence (Q2, SQ4). **We kindly invite the reviewer to consider adjusting the score so that we may have the opportunity to present this work at ICML.**

---

### Official Review · Reviewer_ZHTT · 2026-03-11

**Soundness:** 4
**Presentation:** 3
**Significance:** 3
**Originality:** 3
**Overall Recommendation:** 5
**Confidence:** 3

**Summary:**

The paper proposes ECG-R1, a multimodal reasoning LLM for ECG interpretation that jointly handles ECG time-series signals and ECG images. The key components are: (1) a protocol-guided instruction data generation pipeline that uses deterministic ECG feature extraction plus monograph-derived diagnostic rules to produce structured CoT style interpretations; (2) a decoupled modality architecture with Interleaved Modality Dropout (IMD) that aims to improve robustness and consistency when one modality is missing or token order changes; and (3) reinforcement learning with ECG Diagnostic Evidence Rewards (EDER) that rewards stepwise evidence grounding, in addition to final diagnosis accuracy and format. Extensive experiments show large gains over proprietary, open-source, medical, and ECG-specialized MLLMs on ECG interpretation benchmarks, robustness under modality missing, and cardiologist-rated usefulness.

**Compliance With Llm Reviewing Policy:**

Affirmed.

**Final Justification:**

This is a well-executed and practically meaningful work that addresses an important problem in ECG understanding. The methodology is thoughtfully designed, particularly the protocol-guided CoT generation and the robustness-oriented IMD mechanism, and is supported by comprehensive experiments. Importantly, the authors have provided thorough and convincing rebuttal responses, including additional cardiologist evaluations, quantitative validation of data quality, and analysis of RL distribution effects, which adequately address the main concerns. Overall, the paper is technically solid with clear clinical relevance and makes a valuable contribution to multimodal medical reasoning.

**Key Questions For Authors:**

1. How well do the “top-500 most frequent report texts” used to construct the RL dataset align with the evaluation distribution, and could focusing RL on only the most common reports fail to improve performance on rare but clinically important conditions?

**Limitations:**

Yes. The authors acknowledge the high-stakes nature of ECG interpretation, the research-only intent of their models, and the need for clinical oversight and validation. They could briefly address societal risks, such as misuse by non-experts or underperformance on underrepresented populations.

**Strengths And Weaknesses:**

**Strengths:**

**1. Well-structured methodology and clear presentation:** The design of ECG-R1 follows the general workflow of multimodal reasoning models while incorporating the specific characteristics of ECG data. The paper is well organized, and the figures and tables clearly illustrate the architecture, pipeline, and results.

**2. Well-designed protocol-guided interpretation pipeline:** The protocol-guided workflow is thoughtfully designed. The think reasoning followed by a summary and answer diagnosis provides good clinical interpretability and reduces arbitrary reasoning compared with generic LLM outputs.

**3. IMD:** IMD randomly drops modalities and swaps modality token order during training, exposing the model to different possible test-time input patterns. This helps address sensitivity to missing modalities and cross-modal inconsistencies. The authors also provide a theoretical analysis showing improved robustness and consistency.

**4. Improved architecture:** Compared with GEM’s coupled encoding design, ECG-R1 uses a decoupled architecture where each modality has its own projector and can be used independently, avoiding a fixed alignment bottleneck.

**5. Comprehensive experiments with strong performance:** ECG-R1 outperforms a wide range of proprietary, open-source, medical, and ECG-specific baselines. It achieves higher diagnostic accuracy than GEM and PULSE, improves fine-grained metrics such as ECG feature recognition and evidence-based reasoning, and shows stronger robustness and better cardiologist evaluation results.

**Weaknesses:**
1. Both from a methodological perspective and the qualitative comparison in Figure 5, the high quality of the proposed ECG Protocol-Guided Grounding CoT data appears to be supported. However, to more fully substantiate this claim, some sampling and quantitative evaluation, such as measuring coverage across diagnostic dimensions or rates of medical errors, would provide more objective evidence.

2. Although the other models in Table 1 perform poorly, using only GEM as a baseline in other experiments is still not ideal, especially in the Cardiologist Evaluation. Including a broader set of comparative models （may be at least PULSE）would provide a more comprehensive context for evaluating ECG-R1’s performance and strengthen the reliability of the conclusions.

3. Focusing the RL dataset solely on the top-500 most frequent reports risks neglecting rare but clinically important conditions, creating a distribution shift that may limit or even harm performance on critical cases.

---

> ### Author Rebuttal · Authors · 2026-03-29
>
> We sincerely thank the reviewer for the thorough evaluation and valuable suggestions. Below, we respond to each concern point by point.
>
>
>
> > [W1] Both from a methodological perspective and the qualitative comparison in Figure 5, the high quality of the proposed ECG Protocol-Guided Grounding CoT data appears to be supported. However, to more fully substantiate this claim, some sampling and quantitative evaluation, such as measuring coverage across diagnostic dimensions or rates of medical errors, would provide more objective evidence.
>
> We sincerely thank the reviewer for this suggestion. **To address this concern, we conducted an additional human evaluation, randomly sampling 100 training instances and inviting four licensed cardiologists to evaluate them on three criteria**: Diagnostic Completeness (coverage of clinically relevant abnormalities), Factual Correctness (absence of medical errors), and Reasoning Traceability (whether conclusions are grounded in specific leads and waveform features).
>
> | Metric     | Diagnostic Completeness | Factual Correctness | Reasoning Traceability |
> | ---------- | ----------------------- | ------------------- | ---------------------- |
> | Mean (Std) | 4.61/5 (0.23)           | 4.54/5 (0.23)       | 4.60/5 (0.25)          |
>
> All three metrics score above 4.5/5 with low inter-rater variance, confirming high corpus quality. We will include this analysis in the revised paper.
>
>
>
> > [W2] Although the other models in Table 1 perform poorly, using only GEM as a baseline in other experiments is still not ideal, especially in the Cardiologist Evaluation. Including a broader set of comparative models （may be at least PULSE）would provide a more comprehensive context for evaluating ECG-R1’s performance and strengthen the reliability of the conclusions.
>
> Thanks for your great suggestion, and **we conducted additional cardiologist evaluations with PULSE and GPT-5.1-Instant, using the same 100 test cases and four licensed cardiologists.**
>
> | Model           | Analytical Relevance | Analytical Accuracy | Analytical Completeness | Reasoning Quality | Findings Novelty | Clinical Value  | Overall Satisfaction |
> | --------------- | -------------------- | ------------------- | ----------------------- | ----------------- | ---------------- | --------------- | -------------------- |
> | GPT-5.1-Instant | 2.80 (0.73)          | 2.83 (0.33)         | 2.75 (0.66)             | 2.80 (0.73)       | 2.46 (1.00)      | 2.66 (0.52)     | 2.66 (0.44)          |
> | PULSE           | 3.59 (1.01)          | 3.97 (0.10)         | 3.32 (1.53)             | 3.34 (1.46)       | 2.94 (1.05)      | 3.89 (0.17)     | 3.90 (0.15)          |
> | GEM             | 4.16 (0.18)          | 3.89 (0.04)         | 4.05 (0.15)             | 4.03 (0.14)       | 2.82 (0.82)      | 3.84 (0.19)     | 3.84 (0.11)          |
> | ECG-R1 (Ours)   | **4.55** (0.39)      | **4.34** (0.36)     | **4.44** (0.35)         | **4.49** (0.37)   | **3.25** (0.95)  | **4.38** (0.30) | **4.38** (0.34)      |
>
> PULSE achieves slightly higher Analytical Accuracy than GEM, likely because its more concise outputs reduce the occurrence of factual errors, but at the cost of lower Completeness and Reasoning Quality. ECG-R1 achieves the highest scores on both dimensions. We will include this in the revised paper.
>
>
>
> > [W3 & Q1] Focusing the RL dataset solely on the top-500 most frequent reports risks neglecting rare but clinically important conditions, creating a distribution shift that may limit or even harm performance on critical cases. How well do the “top-500 most frequent report texts” used to construct the RL dataset align with the evaluation distribution, and could focusing RL on only the most common reports fail to improve performance on rare but clinically important conditions?
>
> We greatly appreciate the reviewer for raising this important concern. To address this, we first examined the alignment between the RL training distribution and the evaluation set. The top-500 reports cover 57.9% of the test samples, meaning 42.1% of the evaluation is conducted on reports outside the RL training distribution. We then compared Diagnosis Accuracy between the SFT baseline and the RL-trained model on both groups.
>
> | Group          | SFT Acc (%) | RL Acc (%) | Δ     |
> | -------------- | ----------- | ---------- | ----- |
> | In top-500     | 86.82       | 88.02      | +1.19 |
> | Not in top-500 | 69.03       | 69.68      | +0.65 |
>
> RL yields consistent improvements on both groups, including reports outside the RL training distribution. We attribute this to our EDER reward evaluating stepwise evidence grounding rather than rewarding match to specific report templates. We will discuss this in the future work section of the revised paper and explore RL strategies tailored to rare conditions. We sincerely thank the reviewer for highlighting this point. Finally, we will address the suggested societal risks in the impact statement of the revised paper.

---

> > ### Author Rebuttal · Reviewer_ZHTT · 2026-04-03
> >
> > Thank you to the authors for the detailed response. My concerns have been adequately addressed, and I will maintain my score.

---

> > > ### Author Response · Authors · 2026-04-05
> > >
> > > We sincerely thank the reviewers for their careful reading of our work and for providing constructive feedback. We have carefully considered all comments and made corresponding improvements in the revised manuscript.

---

### Decision · Program_Chairs · 2026-04-30

**Decision:**

Accept (regular)

**Comment:**

This paper addresses an important and practical problem: reliable multimodal ECG interpretation under realistic conditions such as missing modalities and the need for evidence-grounded reasoning. Reviewers agreed that the paper is technically solid, well motivated, and supported by strong empirical results. In particular, the protocol-guided data generation pipeline, the decoupled multimodal design with interleaved modality dropout, and the evidence-rewarded RL stage form a coherent framework that addresses clear failure modes in prior ECG MLLMs.

The main concerns raised during review were about more objective validation of the generated reasoning data, the breadth of comparative evaluation, the effect of focusing RL on frequent report types, evaluator dependence, and the precise scope of some novelty claims. I have read the reviews, rebuttal, and discussion carefully. The rebuttal substantially strengthened the paper by adding cardiologist evaluations against broader baselines, independent evaluator results, quantitative validation of corpus quality, and analysis of the RL distribution issue. While some limitations remain, particularly around rare conditions and measurement supervision, I find the overall contribution strong and well supported.

I therefore recommend accept. In the final version, the authors should slightly sharpen claim calibration and state limitations around rare conditions, measurement accuracy, and research-only use more explicitly.